# Extended Maximal Covering Location and Vehicle Routing Problems in Designing Smartphone Waste Collection Channels: A Case Study of Yogyakarta Province, Indonesia

**Diana Puspita Sari [1,2,*], Nur Aini Masruroh [1,*]** and **Anna Maria Sri Asih [1]**

1   Department of Mechanical and Industrial Engineering, Universitas Gadjah Mada,
    Yogyakarta 55281, Indonesia; amsriasih@ugm.ac.id
2   Department of Industrial Engineering, Diponegoro University, Semarang 50275, Indonesia
*   Correspondence: dianapuspitasari@lecturer.undip.ac.id (D.P.S.); aini@ugm.ac.id (N.A.M.)

**Abstract:** Most people will store smartphone waste or give it to others; this is due to inadequate waste collection facilities in all cities/regencies in Indonesia. In Yogyakarta Province, there is no electronic waste collection facility. Therefore, an e-waste collection network is needed to cover all potential e-waste in the province of Yogyakarta. This study aims to design a collection network to provide easy access to facilities for smartphone users, which includes the number and location of each collection center and the route of transporting smartphone waste to the final disposal site. We proposed an extended maximal covering location problem to determine the number and location of collection centers. Nearest neighbor and tabu search are used in forming transportation routes. The nearest neighbor is used for initial solution search, and tabu search is used for final solution search. The study results indicate that to facilitate all potential smartphone waste with a maximum distance of 11.2 km, the number of collection centers that must be established is 30 units with three pick-up routes. This research is the starting point of the smartphone waste management process, with further study needed for sorting, recycling, repairing, or remanufacturing after the waste has been collected.

**Keywords:** smartphone waste; collection center; extended maximal covering location problem; transportation route; mathematical model; tabu search

## 1. Introduction

Developing countries such as Indonesia currently have the problem of handling large volumes of electronic waste (e-waste) [1]. It is associated with rapid technological and economic developments, leading to the production of a wider selection of electronic products at more affordable prices [2], thereby increasing public consumption and potential for electronic waste. The Global E-waste Monitor 2017 Quantities, Flows, and Resources ranked Indonesia ninth among the global producers of electronic waste, with smartphones being observed to contribute significantly. It is, however, important to note that the use of smartphones started increasing in 2020 due to the emergence of coronavirus, which prompted people to work and learn from home using online platforms. Records show smartphones are the technological devices with the highest consumption rate (70%), followed by laptops and personal computers [3], but there is no appropriate waste management process [4]. This is indicated by the absence of regulations for the collection and transportation of electronic wastes in Indonesia, with those implemented observed to be limited to informal initiatives. This, therefore, led to the low ranking of the country in the waste management system by the United Nations University. This is one of the major differences between Indonesia and developed countries [5]. A previous study also showed that improper handling of waste is dangerous for environmental sustainability [6].

About 80% of the materials composing smartphones can be recycled effectively [7]. Smartphones contain valuable materials, such as gold, silver, and palladium [8]. Metals

in electronic waste, especially smartphones, are present in higher concentrations than in primary ore found in the ground. As an illustration, 300–350 g of secondary gold can be extracted from one ton of smartphones, while every ton of soil in ordinary gold mines only produces 5 g of primary gold [9]. Resource extraction from e-waste is more economical than extracting metal ores from the ground [10]. Thus, smartphone recycling is done because the economic benefits outweigh the costs [11]. Proper management of e-waste is necessary to reduce the problem of metal scarcity [8]. The potential for smartphone waste in Indonesia is quite significant. The total population of Indonesia in 2020 was 270,203,917 people [12]. If 63.53% are smartphone users [13], then the total number of smartphone users is 171,660,549 people. With an average smartphone lifetime of 4.7 years [14], these users produce 36,523,521 units of smartphone waste per year. When this waste is appropriately managed, in addition to minimizing the environmental impact, it can also provide significant economic benefits by producing 5.48–6.39 tons of secondary gold and saving natural resources.

However, so far, the amount of secondary metal recovered through e-waste recycling has been limited [15]; this is due to the limited supply of e-waste. A preliminary study conducted on smartphone users in Indonesia showed that 59% save non-functioning smartphones; 21% dispose of them; and the rest give them out to other people, sell them, and use them in other ways. This is because the public does not know what to do with these items. Meanwhile, Yogyakarta is one of the barometer provinces in Indonesia with an improper electronic waste management system through the formal channel. According to previous studies, government drivers are the factor with the most influence on consumers' intentions to participate in smartphone waste collection programs, followed by facility accessibility [16]. This means that the government needs to develop and implement a formal e-waste management system, starting with the e-waste collection process. One of the alternative electronic waste collection programs applicable to Indonesia is the use of a dropbox [17], but Yogyakarta Province does not currently have any collection points for smartphone waste. Therefore, there is the need to provide a convenient collection channel for the consumers, which is expected to be a major starting point for a formal channel to waste management in the area.

This study aimed to design a collection channel by determining the number and location of the collection center facilities followed by a transportation route from the collection center to the final disposal site. Facility location is related to the finding of a solution that covers customers using a number of facilities. It is, however, important to note that covering problems are fundamental facility location problems [18], which are often categorized as location set covering problems (LSCPs) and maximal covering location problems (MCLPs). The classic MCLP involves looking for the location of several facilities on the network in such a way that the population covered can be maximized [19]. Church and ReVelle first introduced this model in 1973 at a North American Regional Science Council [20]. The purpose was to maximize the demand covered by a particular service distance by placing a certain number of distribution facilities [21]. Therefore, customers or clients are declared covered when they are within a certain coverage distance from at least one facility [22]. The model is also important in the decision-making of the supply chain process, making it relatively important for practical use [23].

Several previous studies have used MCLP to design models and approaches in determining locations. MCLP is used in both the public and private sectors. In the public sector, it has been applied to determine the spread of an ambulance in emergency services [24], the location of emergency warning sirens [25], the location of medical equipment supply centers [26], the location of treatment centers in the event of a disease outbreak [27], appropriate locations for shelters for those temporarily displaced by floods [28], and the location of a waste cooking oil collection center [29]. Its use in the private sector involves determining the location of bank branches [30]. Several researchers have developed MCLPs. For example, Davari et al. [31] developed a MCLP with fuzzy travel times; Arana-Jiménez et al. [32] developed a fuzzy MCLP; Vatsa and Jayaswal ([33,34]) modeled a capacitated

multiperiod MCLP with server uncertainty; and Cordeau et al. [9] introduced the MCLP algorithm to determine a subset of facilities, maximizing customer requests by considering budget constraints. A continuous MCLP was also developed by Yang et al. [35] to optimize a continuous location of the cellular network's communication centers for natural disaster rescue. ReVelle et al. [36] also solved the MCLP with heuristic concentration which is used to determine a prominent case solution to maximum coverage locations with a high coverage percentage. Ibarra-Rojas et al. [37] developed a MCLP with accessibility indicators for when facilities have limited service areas, while Alizadeh and Nishi [38] used the hybrid covering location problem for strategic and tactical decisions. Alizadeh and Nishi [39] also developed a multiperiod maximal coverage location problem with different facility configurations as an extension of the classic MCLP. Zhang et al. [40] addressed the issue of locating multimodal facilities in emergency medical rescue.

The classical MCLP is used to determine the minimum number of facilities to maximize the demand covered by a given service distance. The model does not consider costs; it assumes that the number of facilities is minimal and that the investment costs are also minimal. Because each alternative location is assumed to have the same investment costs, it is necessary to develop a model that considers the difference in investment costs between potential locations. In this study, the collection center to be built is an intermediary facility, so it is also necessary to consider transportation costs to the final facility. Therefore, in this study, we develop the MCLP method by considering the investment and transportation costs, hereinafter referred to as an extended maximal covering location problem (e-MCLP). With this development, in addition to minimizing the number of facilities, it will also minimize total costs, including investment costs and transportation costs. Thus, the developed model is expected to provide an affordable facility location to consumers with a minimum total investment and transportation costs from the collection center to the final disposal facility.

The selection of the number and location of the collection centers was followed by the transportation route scheduling plan from the collection center to the final disposal site to determine the optimal route for efficient product distribution. It is defined as the route with the shortest distance and is considered important due to its ability to reduce transportation costs [41]. The vehicle route optimization problem, however, is known as the vehicle routing problem (VRP), which was introduced by Dantzig and Ramser in 1959 to solve the problem of gasoline distribution [42]. VRP is a common discrete optimization problem in transportation and logistics [43]. It is generally an integral part of the vehicle route with the exact delivery location visited once while all the routes start and end at the warehouse [44]. VRP focuses on the distribution of goods from the company's depot to customers and aims to minimize global transportation costs related to distance, fixed costs associated with vehicles and balance routes, and the number of vehicles required to serve consumers [45]. There are three methods of solving VRPs: the exact, heuristic, and metaheuristic methods [46]. However, the exact method is not applicable to a problem with a large input size and a limited time.

The methods used in this study include the heuristic and metaheuristic methods. The heuristic method involved the application of the nearest neighbor (NN) method, which has been widely used to solve VRP. Solomon introduced it in 1987 based on the idea of visiting the closest location from every other location visited [47], and it has been observed to be significantly better and to have more realistic performance in route formation than other methods [48]. This led to its wide application in solving the traveling salesman problem [49], determining routes from one city to another [50], designing waste transportation routes [51], and minimizing travel time and fuel consumption for transportation of agricultural products [52]. The nearest neighbor method is quite effective in its application due to its ability to look for consumers based on the closest distance from the vehicle's last location. It is, important to note that the nearest neighbor method produces the route with the shortest distance compared to other heuristic methods [41]. It is also easy to implement

and execute the algorithm, but it does not guarantee the best resulting solution [53], so in this study, nearest neighbor was used to determine the initial solution.

This research applied the tabu search (TS) method, an algorithm considered to have the ability to produce an optimal solution. It was first introduced by Glover [54] based on the idea that allowing uphill motion helps to prevent the solution from becoming stuck in local optimal conditions [55]. The strength of this method lies in its flexible memory structure [54]. This makes its solutions very similar every time it is applied and makes it better than the other methods, such as simulated annealing and genetic algorithm [54]. Several studies have used tabu searches to solve VRP [56], classical VRP, periodic VRP, multidepot VRP, site-dependent VRP [57], heterogeneous fleet VRP [58], VRP with discrete split deliveries and pickups [59], multicompartment VRP [60], heterogeneous multitype fleet VRP with time windows and an incompatible loading constraint [61], multidepot open VRP [62], VRP with cross docks and split deliveries [63], VRP with private fleet and common carrier [57], time-dependent VRP with time windows on a road network [64], consistent VRP [65], and heterogeneous VRP on a multigraph [66]. Shi et al. [67] also used the heuristic solution method for the problem of multidepot vehicle routing-based waste collection and compared the results with the tabu search. Khan et al. [68] presented a sustainable closed-loop supply chain framework that uses a metaheuristic approach, tabu search, and simulated annealing. Tebaldi et al. [69] determined the best route to visit a set of customers, considering vehicle capacity and time constraints. This result underlies the use of the nearest neighbor approach to obtain an initial solution and the use of the metaheuristic tabu search approach to determine the final solution.

## 2. Materials and Methods

This research was conducted in two main stages: determining the number and location of collection centers and determining the smartphone waste transportation route. The location, number, and capacity of collection centers were determined by developing a maximal covering location problem hereinafter referred to as the extended maximal covering location problem (e-MCLP). The focus of the MCLP is to minimize the number of facilities while ensuring all consumers are covered, but the e-MCLP was developed to consider the costs involved. The model's objective was, therefore, to minimize the total costs, including those associated with investment and transportation from the collection facility to the final disposal site. The costs associated with collecting smartphones are not as high as those for other large volumes of e-waste, but the developed model can be used for other types of waste. The reason for choosing this type of waste is because it has a higher economic value (containing precious metals such as gold, silver, and palladium) than others, with components that allow up to 80% recycling and a large potential for smartphone waste. Meanwhile, for now, informal actors dominate the practice of recycling smartphone waste, which harms the environment. The low collection cost and high economic and environmental benefits are expected to motivate the government to implement the proposed scenario.

The development scenario involves two levels of collection center (CC) facilities, namely the primary collection center (PCC) and the secondary collection center (SCC). Consumers collect their waste at PCC. Instead, local governments carry out transportation from PCC to SCC. Transportation routes are needed in this study because smartphones are products with small volumes, so the capacity of the collection center is not as large as vehicle capacity. If one trip only picks up from one PCC, it becomes inefficient because the vehicle's utility is low, and transportation costs will be higher due to many trips being needed. For this reason, it is necessary to consider the route determination in this study. Routing is expected to increase vehicle utilization and save transportation costs. The output of determining the transportation route is expected to be an input for local governments to schedule waste collection.

Yogyakarta, one of the provinces in Indonesia, is located on Java Island and has an area of 3178.79 km$^2$. It has a municipality and four regencies: Yogyakarta city and

Gunung Kidul, Bantul, Sleman, and Kulon Progo regencies, with respective areas of 32.5, 1485.36, 506.85, 574.82, and 579.26 km$^2$. These areas contain 14, 18, 17, 17, and 12 districts, respectively [70], as indicated in Appendix A, for a total of 78 districts. These districts were used as candidates for primary collection centers (PCCs) in this study. The parameters used as input in the mathematical model include the distance between the PCCs, the distance expected by consumers, and the distance from PCC to SCC.

Yogyakarta Province currently has 3 locations serving as final disposal sites (TPAs). The first is the Regional TPA, commonly called Piyungan TPA in Ngablak, Sitimulyo Village, Piyungan District, Bantul Regency. It is an integrated waste disposal site created to serve Yogyakarta City, Bantul Regency, and Sleman Regency [71]. The second location is Wonosari TPA in Wukirsari, Baleharjo, Wonosari, Gunung Kidul Regency, and the third is the Banyuroto TPA in Dlingo, Banyuroto, Nanggulan, Kulon Progo Regency. The Piyungan TPA has the largest capacity and most strategic location among the three, and this makes it suitable to be used as the secondary collection center (SCC). The candidates for the PCCs are district offices, which means the distances between PCCs are the same as those between district offices, and the distance from the PCC to SCC is the distance from the district office to the TPA Piyungan.

The PCC is provided by the government for consumers in the form of a dropbox, while SCC is a waste collection point for all the PCCs in a province. For this research, one SCC was located at the final disposal site in one province while the PCCs were built at the minimum number required to minimize investment costs incurred but with the ability to reach all consumers. Further, a survey conducted on smartphone users, with a total of 325 valid questionnaires, showed the consumers are willing to bring their smartphone waste to a collection facility with a maximum distance of 11.2 km. This means the PCC to be established is based on the number of districts to accommodate the interests of the consumers. Meanwhile, the PCC with the closest distance to the SCC was selected for this research to accommodate government interests by minimizing transportation costs. The PCC is located in the district office, a government-owned facility, and this means it does not require large investment costs since there is no need to procure land and a building, as only the dropbox needs to be prepared. This collection center has the capacity to accommodate all the smartphone waste supplies in the area due to the small product volume. It is important to determine the transportation route to optimize vehicle utility due to the relatively small volume of waste.

The location and capacity of the PCC were used to determine the transportation routes by joining the nearest neighbor approach and the tabu search model (NN-TB). The application of the NN was initiated from the starting point, which is the depot/SCC, and directed towards the PCC with the closest distance, which has not been visited due to several restrictions. The solution obtained at this stage is limited to determining the best route and the consumers to be served next based on the nearest point to the vehicle's last location [72]. It has been previously stated that the nearest neighbor algorithm is easy to implement and execute but does not guarantee the maximum resulting solution [53], and this was the reason it was used in this study to determine only the initial solution. Afterward, the tabu search method was used to search for the optimal route. The metaheuristic method is usually applied to solve combinatorial optimization problems, where the combinations are usually used to calculate the number of exchanges to be made in each iteration [73]. The tabu search algorithm is also a mathematical optimization method that guides the iterative search for solutions by providing tabu status for solutions found [74].

### 2.1. Collection Center Determination Steps

The parameters used as input in the mathematical model are the distance between PCC candidates, the distance expected by consumers, and the PCC candidate's distance to the final disposal site or secondary collection center (SCC). The distance between PCC candidates and distances between each PCC and SCC were based on Google Maps. The distance matrices between PCC candidates and from the PCC candidates to the SCC are

shown in Appendix B. The distance value is essential to determine the number and location of PCCs to be built in the area.

The notation used in the mathematical model of e-MCLP is as follows:

| | |
|---|---|
| $Z$ | Total cost |
| $IC$ | Investment cost |
| $TC$ | Transportation cost |
| $m$ | The number of the district ($m = 1, 2, \dots, \lvert m \rvert$) |
| $k$ | The number of SCC ($k = 1$) |
| $X_i$ | Supply point $i$ |
| $X_j$ | Point $j$ is selected or not as a PCC |
| $X_j$ | $\begin{cases} 1, & j \text{ become } CCP \; \forall \, j \in V \\ 0, & \text{otherwise} \end{cases}$ |
| $a_{ij}$ | Distance requirements (fulfilled or not) |
| $a_{ij}$ | $\begin{cases} 1, & \text{distance } i \text{ to } j \leq D \; \forall \, i, j \in V \\ 0, & \text{otherwise} \end{cases}$ |
| $D$ | Range (km) |
| $Q_j$ | The capacity of PCC at point $j$ |
| $Y_i$ | Coverage of smartphone waste supply at point $i$ (covered or not) |
| $Y_i$ | $\begin{cases} 1, & \text{the point is covered in the PCC at point } j \; \forall \, i \in V \\ 0, & \text{otherwise} \end{cases}$ |
| $S_i$ | Supply of smartphone waste at point $i$ |

The basic model was developed from the MCLP [75] in the form of e-MCLP, and its functional objective was to minimize the total cost of the number of facilities to be established within the range wanted by the consumer, as shown in Equation (1). The costs considered include those associated with the investment and transportation from PCC to SCC. Furthermore, the PCC was established in a district office, a government facility, which means there was no need to invest money in land acquisition. Therefore, the only investment needed was the procurement of dropbox, and the value is the same for all candidate locations. It is important to note that the PCC locations selected were those with the lowest investment costs and closer to the SCC. The decision variable $X_j$ has a value of 1 or 0, where a value of 1 indicates the point $j$ is selected as a PCC and a value of 0 indicates the point $j$ is not selected as a PC. Dropbox procurement costs are USD 350.37 (USD 1 is equivalent to IDR 14,270.75), and the dropbox service life is 5 years; using the straight-line depreciation method, the annual depreciation cost is USD 70.07 per dropbox. Thus, the investment cost per year is USD 70.07 per dropbox. The vehicle's fuel consumption is 10 km/L at USD 0.67 per liter; therefore the transportation cost is USD 0.067 per kilometer.

Equations (2)–(6) are constraint functions. Equation (2) is a limiting function that requires $a_{ij}X_j$ to be 1, and this means a minimum of one PCC needs to be established within the range of the consumers' point. Meanwhile, Equations (3)–(5) state that $X_j$, $a_{ij}$, and $Y_i$ are binary, while Equation (6) states that the PCC capacity at point $j$ is the accumulation of the waste supply multiplication at point $i$ by 1 or 0, where 1 means the waste supply at point $i$ is covered and 0 means it is not covered. Smartphones are, however, usually in small volume and not too large a supply due to the estimation of lifespan at two years. Therefore, the PCC capacity value used in this research is 1, which indicates that the entire waste supply was accommodated.

$$Min \; Z = \sum_{j=1}^{m} IC_j X_j + \sum_{j=1}^{m} \sum_{k}^{1} TC_{jk} X_j \tag{1}$$

$$\sum_{i,j=1}^{m} a_{ij} X_i \geq 1 \tag{2}$$

$$X_j \in [0,1] \; \forall \, j \in V \tag{3}$$

$$a_{ij} \in [0,1] \; \forall \, i, j \in V \tag{4}$$

$$Y_i \in [0,1] \; \forall \, i \in V \tag{5}$$

$$Q_j = \sum_{i=1}^{m} Y_i S_i, \text{ for every } j \tag{6}$$

*2.2. Steps to Determine the Transportation Route*

The method used in this research was the nearest neighbor and tabu search (NN-TS) method, where the results obtained from the nearest neighbor were used as input in the tabu search. It is important to note that the tabu search was initiated by approaching a local minimum and noting recent movements in a tabu list that forms an adaptive memory to explore better solutions, with its size indicating the degree of diversification and intensification [76]. The mathematical model was, however, first determined before the calculations, and this was based on several assumptions and limitations, which include the following: (1) the vehicle has enough capacity to accommodate smartphone waste; (2) the distance from location $j(a)$ to $j(b)$ is the same as the distance from location $j(b)$ to $j(a)$ due to symmetry; (3) collection activities to PCCs start from 08:00–16:00 WIB with a rest time of 1 h, and this means the planning time horizon for a day is 7 h; (4) one vehicle visits more than one PCC but each PCC is only visited by one vehicle; (5) the average vehicle speed is 45 km/h; (6) the loading time at a PCC is 10 min; (7) the unloading and administration time at the SCC is 30 min. The notation used in the mathematical model of VRP is as follows:

| | |
|---|---|
| $V$ | The set of all vertices with 0 is a SCC $\{0, 1, 2, \dots, v\}$ |
| $P$ | The set of PCC $\{1, 2, \dots, p\}$ |
| $E$ | The set of directed ribs $\{(j(a),j(b)) \mid j(a),j(b) \in V, j(a) \neq j(b)\}$ |
| $T$ | The set of trip $\{1, 2, \dots, t\}$ |
| $C$ | Vehicles $\{1, 2, \dots, c\}$ |
| $J$ | Total mileage (km) |
| $D_{j(a)j(b)}$ | Distance from PCC at point $j(a)$ to $j(b)$ (km) |
| $X_{j(a)j(b)c}^t$ | There is a trip from PCC at point $j(a)$ to $j(b)$ on trip $t$ or not |
| $X_{j(a)j(b)c}^t$ | $\begin{cases} 1, \text{ vehicle } c \text{ travels from point } j(a) \text{ to } j(b) \text{ on the trip } t \\ 0, \text{ otherwise} \end{cases}$ |
| $d_{j(a)}$ | PCC capacity at PCC at point $j(a)$ |
| $Q$ | Load capacity on a route |
| $T_{j(a)j(b)}$ | Travel time from PCC at point $j(a)$ to $j(b)$ |
| $S_c^t$ | Service time (loading–unloading) |
| $Y_{j(a)c}^t$ | There is a load on PCC at point $j(a)$ carried by vehicle $c$ on trip $t$ or not |
| $Y_{j(a)c}^t$ | $\begin{cases} 1, \text{ there is a load at point } j(a) \text{ carried by vehicle } c \text{ on the trip } t \\ 0, \text{ otherwise} \end{cases}$ |

The objective function of the VRP mathematical model is to minimize the total distance traveled from the route as shown in Equation (7). The decision variable $X_{j(a)j(b)c}^t$ has a value of 1 or 0; 1 indicates the selected route when vehicle $c$ travels from PCC at point $j(a)$ to $j(b)$ on the trip t, and 0 indicates when the situation is otherwise. Equations (8)–(14) are constraint functions, with Equations (8) and (9) used to show that the route starts from and returns to SCC. Equations (10) and (11) state that each PCC is served exactly once on one route. Hereinafter, the vehicle's load capacity on a trip is the accumulation of the PCC capacity served, and its maximum capacity is not exceeded, as shown in Equation (12). This is because the supply is not large and the product volume is small, which allows the vehicle to carry the entire supply of smartphone consumers at once. Meanwhile, Equation (13) shows the vehicles going to the SCC to unload. However, the route completion time was calculated from the vehicle's total time plus the service time, which is loading–unloading time, and observed not to have exceeded the planning time horizon in a day, which is 7 h, as shown in Equations (14) and (15).

$$Min\ Z = \sum_{j(a) \in V} \sum_{j(b) \in V} \sum_{t \in T} \sum_{c \in C} C_{j(a)j(b)} X_{j(a)j(b)}^t \tag{7}$$

$$\sum_{j(b) \in P} X_{0j(b)c}^t = 1 \forall c \in C \tag{8}$$

$$\sum_{j(b)\in P} X^t_{j(a)0c} = 1 \forall c \in C \tag{9}$$

$$\sum_{j(b)\in P} \sum_{t\in T} \sum_{c\in C} X^t_{j(a)j(b)c} \forall j(a) \in P, j(b) \neq j(a) \tag{10}$$

$$\sum_{j(b)\in P} \sum_{t\in T} \sum_{c\in C} X^t_{j(a)j(b)c} \forall j(b) \in P, j(a) \neq j(b) \tag{11}$$

$$Q = \sum_{j(a)\in P} d_{j(a)} Y^t_{j(a)c} \forall t \in T, \forall c \in C \tag{12}$$

$$\sum_{j(a)\in P} X^t_{j(a)0c} = 1 \forall t \in T, \forall c \in C \tag{13}$$

$$CT = \sum_{j(a)\in V} \sum_{j(b)\in V} \sum_{t\in T} T_{j(a)j(b)} X^t_{j(a)j(b)} + \sum_{t\in T} S^t_c, \forall c \in C \tag{14}$$

$$CT \leq 7 \tag{15}$$

The steps to determine the initial solution using the nearest neighbor method [72] are as follows:

a.   Select the center point as the starting point of transport, which is the SCC in this study.
b.   Determine the point with the smallest distance from SCC and move to the PCC point.
c.   The last point visited is the starting point; therefore, determine the point with the closest distance from the point.
d.   Repeat the process until the vehicle does not have sufficient capacity for transportation; but because there is always enough capacity of the vehicle used in this research, the repetition is conducted until it meets the planning time horizon for a day but does not exceed it.
e.   Drag this point to a line which is called a route with the working hours used as a constraint to form a freight route.

The Tabu search algorithm used in this study is based on [77,78] and includes the following steps:

a.   Determine solution representation. This is a sequence of nodes where each is only visible once in the sequence. These nodes represent PCC and SCC.
b.   Formulate initial solution formation, S.
c.   Determine the neighborhood solution. This is an alternative solution obtained by moving the nodes such that each move produces a neighborhood solution and the number of solutions is calculated using the following Equation (16):

$$C_{(n,2)} = \frac{n!}{2!(n-2)!} \tag{16}$$

where $n$ is the number of PCCs visited in a route.

d.   Create a tabu list. This list contains the moved attribute previously found, and its length increases with the size of the issue and also corresponds to the number of PCCs to be visited.
e.   Find the best solution, S*.
f.   Fix the tabu list.
g.   Determine aspiration criteria. This is a method of overturning the tabu status.
h.   Determine termination criteria. These are used after all predetermined iterations have been fulfilled. The number of iterations selected is the same as the number of points visited because the maximum number of iterations is the same as the length of the tabu list [79].

## 3. Results

### 3.1. Number and Location of Collection Centers

The number and location of the PCCs were determined using the e-MCLP method. The solver software was used to determine the optimal solution. The calculations showed that 30 PCCs are to be built as shown in Figures 1 and 2 with a distribution of 1 unit in Yogyakarta city (Y6), 13 units in Gunung Kidul Regency (G1, G2, G4, G5, G6, G7, G8, G9,

G13, G14, G15, G16, and G17), 6 units in Bantul Regency (B4, B10, B12, B13, B15, and B16), 6 units in Sleman Regency (S2, S6, S11, S12, S13, and S17), and 4 units in Kulon Progo Regency (K5, K8, K10, and K12). The selected PCC numbers and locations are shown in Appendix C.

| Y | Y1 | Y2 | Y3 | Y4 | Y5 | Y6 | Y7 | Y8 | Y9 | Y10 | Y11 | Y12 | Y13 | Y14 | Y15 | | | |
|---|----|----|----|----|----|----|----|----|----|-----|-----|-----|-----|-----|-----|----|----|----|
| G | G1 | G2 | G3 | G4 | G5 | G6 | G7 | G8 | G9 | G10 | G11 | G12 | G13 | G14 | G15 | G16 | G17 | G18 |
| B | B1 | B2 | B3 | B4 | B5 | B6 | B7 | B8 | B9 | B10 | B11 | B12 | B13 | B14 | B15 | B16 | B17 | |
| S | S1 | S2 | S3 | S4 | S5 | S6 | S7 | S8 | S9 | S10 | S11 | S12 | S13 | S14 | S15 | S16 | S17 | |
| K | K1 | K2 | K3 | K4 | K5 | K6 | K7 | K8 | K9 | K10 | K11 | K12 | | | | | | |

**Legend**

Selected

Not selected

**Figure 1.** The output of the solver.

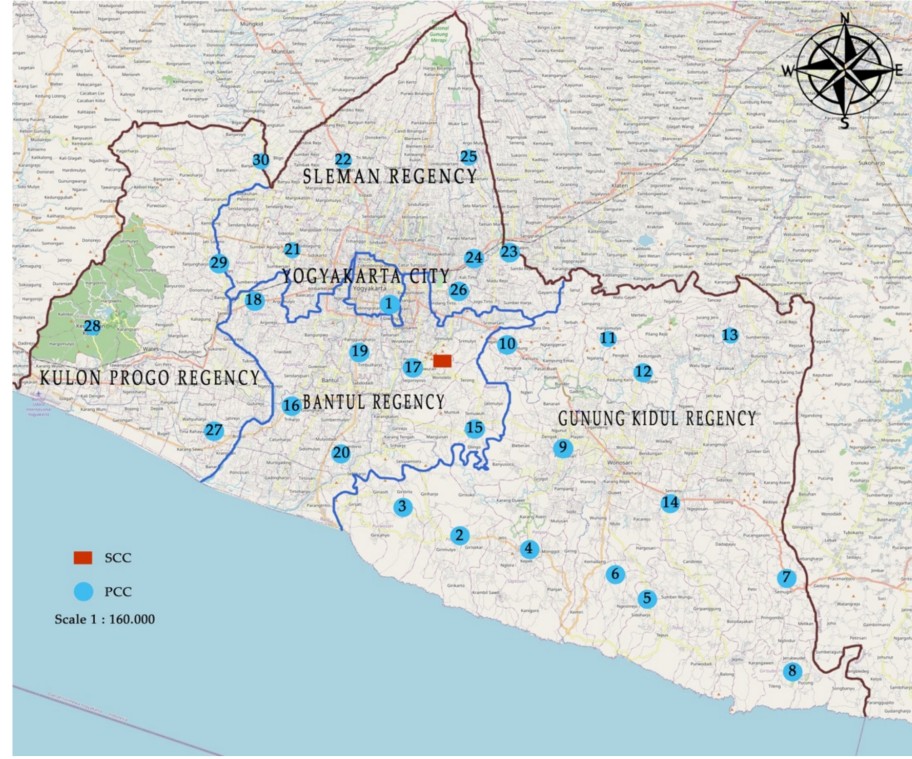

**Figure 2.** Locatio ns of selected PCCs.

The location chosen for the PCC construction in the city of Yogyakarta is Kotagede district. Those selected in Gunung Kidul Regency include the districts of Panggang, Purwosari, Sapto-sari, Tepus, Tanjungsari, Rongkop, Girisubo, Playen, Patuk, Gedangsari, Nglipar, Ngawen, and Semanu. Locations in Bantul Regency include the districts of Dlingo, Pandak, Pleret, Sedayu, Sewon, and Pundon while those in Sleman Regency are Godean, Sleman, Pambanan, Kalasan, Cangkringan, and Berbah districts. Those selected in Kulon Progo Regency included Lendah, Kokap, Nanggulan, and Kalibawang. The total accumulated distance from 30 selected PCCs to SCC is 948.7 km with a transportation cost of USD 126.31 for the scenario of collecting each pick-up from CCS to only one PCC. Since there are 30 CCPs established, the required annual investment cost is USD 2102.2. The total population of Yogyakarta in 2020 was 3,842,932 people; if 63.53% are smartphone users, the smartphone lifetime is 4.7 years, and the average weight of a smartphone is 0.5 kg, then

the average smartphone waste supply in Yogyakarta per year is approximately 305,176 kg. Considering the capacity of the dropbox, pick-up should be done once a week, If the waste collection is done once a week, then the total cost required is USD 8165.07 per year

### *3.2. The Results of Scheduling the Transportation Route*

The nearest neighbor method's search for initial solutions started with the 7 h obtained for planning horizon time, a loading time of 10 min for each PCC, and 30 min of unloading and administration time at SCC. This was followed by the determination of the depot as the starting location, which is the SCC. The vehicle has the capacity to accommodate the entire PCC because the supply is not large and the product volume is small; therefore, the planning time horizon was considered. The next step was the determination of the PCC with the closest distance, and this was discovered to be Pleret PCC, which has a distance of 4.3 km from the SCC. The distance matrices between the selected PCCs and from the selected PCC to the SCC are shown in Appendix D. It is important to note that the retrieval process was continued to the next PCC when the completion time (CT) was less than or equal to the planning time horizon but canceled when the completion time was greater than the planning time horizon. Furthermore, the next PCC was determined based on the closest distance with the initial steps implemented when it was discovered not to have been served. It is also important to point out that just one type of vehicle was used.

The number of trips or tours required to make the collection was calculated to be 3 with a total distance of 659.1 km, travel time of 14.65 h, and a completion time of 21.16 h, as shown in the sequence presented in Table 1. Route 1 had 13 PCCs with a total distance of 193.9 km, travel time of 4.31 h, and completion time of 6.98 h. Route 2 had 10 PCCs with a total distance of 198.4 km, travel time of 4.41 h, and completion time of 6.58 h. Route 3 had 7 PCCs with a total distance of 266.8 km, travel time of 5.93 h, and completion time of 7.6 h. It was discovered that Route 3 has a longer travel time than the planning time horizon, and it was used as an initial solution in the tabu search method with the expectation that it will improve and provide shorter distances and times for the optimal solution.

**Table 1.** The initial solution results using nearest neighbor.

| Route | Picking Sequence |
|---|---|
| 1 | SCC→$PCC_{17}$→$PCC_1$→$PCC_{26}$→$PCC_{24}$→$PCC_{23}$→$PCC_{10}$→$PCC_9$→$PCC_{15}$→$PCC_{19}$→ $PCC_{16}$→$PCC_{27}$→$PCC_{18}$→$PCC_{29}$→SCC |
| 2 | SCC→$PCC_{20}$→$PCC_3$→$PCC_2$→$PCC_4$→$PCC_6$→$PCC_5$→$PCC_8$→$PCC_7$→$PCC_{14}$→$PCC_{12}$→ SCC |
| 3 | SCC→$PCC_{21}$→$PCC_{22}$→$PCC_{25}$→$PCC_{11}$→$PCC_{13}$→$PCC_{30}$→$PCC_{28}$→SCC |

The tabu search method was applied using the initial solution calculated from the nearest neighbor method. Route 1 was found to be SCC–Pleret PCC–Kotagede PCC–Sewon PCC–Pandak PCC–Bambanglipuro PCC–Sedayu PCC–Dlingo PCC–Playen PCC–Patuk PCC–Ngglipar PCC–Ngawen PCC–SCC. This was followed by the input of the number of elements to be searched, which was found to be in accordance with the points to be visited, i.e., 11 PCCs. The number of neighborhood solutions was later determined using Equation (14), and 55 lines were recorded. Furthermore, the tabu list length was discovered to be in line with the number of PCCs to be visited, which was 11 customer locations. This was followed by the maximum number of iterations, which was recorded to be 11 iterations in line with the number of PCCs. These steps were repeated for the other routes, and the determination of the best route produced three routes with a total distance of 602.2 km, a travel time of 13.4 h, and a completion time of 19.89 h. The time for a shipment was found to be 3 days. Furthermore, the best sequences for Routes 1, 2, and 3 had total distances of 178.3, 198.3, and 224.5 km; travel times of 3.98, 4.41, and 5.01 h; and completion times of 6.63, 6.58, and 6.68 h, respectively, as shown in Figure 3 and Table 2.

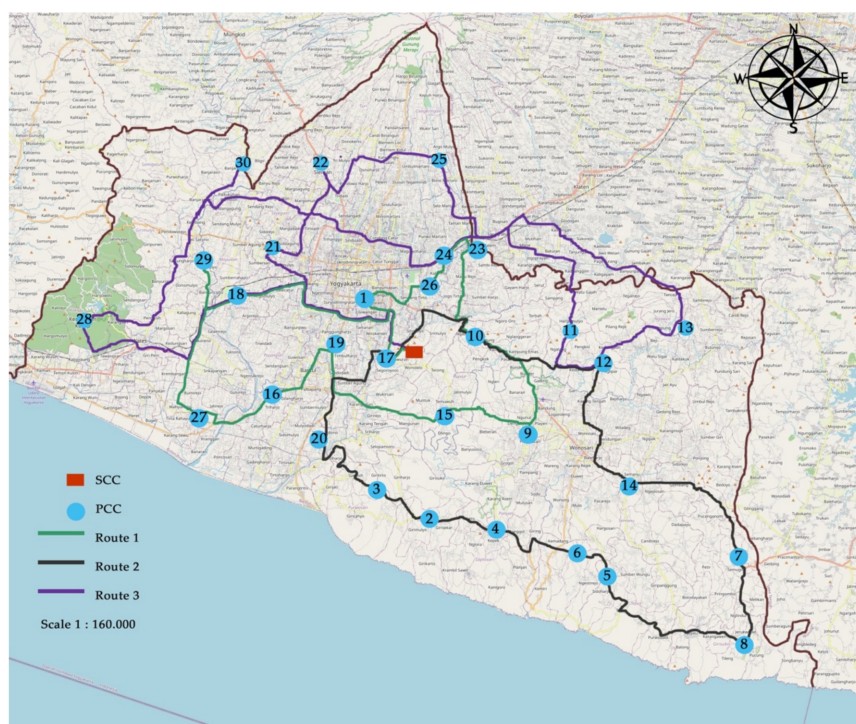

**Figure 3.** Collection routes.

**Table 2.** The final solution results using tabu search.

| Route | Picking Sequence |
|---|---|
| 1 | SCC→PCC$_1$→PCC$_{19}$→PCC$_{18}$→PCC$_{29}$→PCC$_{27}$→PCC$_{16}$→PCC$_{17}$→PCC$_{15}$→PCC$_9$→PCC$_{10}$→PCC$_{23}$→PCC$_{24}$→PCC$_{26}$→SCC |
| 2 | SCC→PCC$_{12}$→PCC$_{14}$→PCC$_7$→PCC$_8$→PCC$_5$→PCC$_6$→PCC$_4$→PCC$_2$→PCC$_3$→PCC$_{20}$→SCC |
| 3 | SCC→PCC$_{13}$→PCC$_{11}$→PCC$_{25}$→PCC$_{22}$→PCC$_{21}$→PCC$_{30}$→PCC$_{28}$→SCC |

## 4. Discussion

The results showed that the city/regency with the fewest PCCs is Yogyakarta city due to the short distance between its districts, with the one PCC established in Kota Gede district being found to have the ability to reach 13 other districts. The farthest is the Tegalrejo district, which is 9.5 km away, and this is also considered to be within the distance desired by the consumers. Meanwhile, most of the PCCs were built in Gunung Kidul Regency due to its large area relative to the other cities and regencies, and this caused quite a long distance between the districts. The area is 47% of the total area of Yogyakarta Province, as shown in Figure 2. Therefore, there is a need to build 13 PCCs in the existing 18 districts to cover all consumers, and the remaining 5 will be accessible because they are less than 10 km from the built locations. For example, Playen PCC covers Paliyan District while Semanu PCC covers Ponjong and Karangmojo districts. It is also possible for the waste from Wonosari District to be transported to Playen or Semanu PCC, while Ngawen PCC covers the Semin district.

This problem, if solved using MCLP as done by Church and Davis [70], Murray [17], Boonmee et al. [18], and Hartini et al. [26] to minimize the number of collection centers that must be built, results in the same number of PCCs that must be built as with e-MCLP, namely 30 PCCs; the number of PCCs established in each city/regency is the same, but there are several different locations. The comparison of selected CCP locations from each method is shown in Figure 4. The different locations are Kotagede, Semanu, Berbah, and Kalibawang when using the e-MCLP method. When using the MCLP method, the selected locations are Gondomanan, Ponjong, Minggir, and Samigaluh, as shown in Figure 4. The difference between the four locations will have implications for saving transportation costs

from PCC to SCC because of the shorter distance, while the investment costs, in this case, are the same for each selected PCC. Comparison of the distance from PCC to SCC between the two methods is shown in Table 3

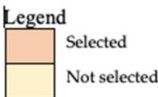

**Figure 4.** Comparison of the solver output between e-MCLP and MCLP methods.

**Table 3.** Distance comparison between e-MCLP and MCLP results.

| MCLP | | e-MCLP | | The Difference in Distance (km) |
| --- | --- | --- | --- | --- |
| **Selected PCC** | **Distance to SCC (km)** | **Selected PCC** | **Distance to SCC (km)** | |
| Gondomanan (Y4) | 13.1 | Kotagede (Y6) | 9.0 | 4.1 |
| Ponjong (G10) | 45.8 | Semanu (G9) | 39.9 | 5.9 |
| Minggir (S3) | 34.0 | Berbah (S13) | 9.1 | 24.9 |
| Samigaluh (K11) | 46.6 | Kalibawang (K12) | 40.6 | 6.0 |
| Total of difference in distance | | | | 40.9 |

Table 4 shows that the distance between the selected PCCs and SCC is shorter in e-MCLP than MCLP. This indicates the numbers and locations calculated using the two approaches were able to accommodate the range expected by the consumers, but e-MCLP considered the investment costs and the distance between the PCC and SCC, unlike the MCLP. Therefore, MCLP provided a greater total PCC to SCC distance, which is indicated by 989.6 km with transportation costs of USD 131.76 when the collection is at only one PCC and the total cost required per year is USD 8426.27. Meanwhile, e-MCLP provided a shorter total distance of 40.9 km with 4.13% savings in transportation costs at USD 261.6 per year.

**Table 4.** Comparison of nearest neighbor and tabu search results.

| Route | Nearest Neighbor | | | Tabu Search | | |
| --- | --- | --- | --- | --- | --- | --- |
| | **D (km)** | **TT (hours)** | **CT (hours)** | **D (km)** | **TT (hours)** | **CT (hours)** |
| 1 | 193.9 | 4.31 | 6.98 | 178.3 | 3.98 | 6.63 |
| 2 | 198.4 | 4.41 | 6.58 | 198.4 | 4.41 | 6.58 |
| 3 | 266.8 | 5.93 | 7.60 | 225.5 | 5.01 | 6.68 |
| Total | 659.1 | 14.65 | 21.16 | 602.2 | 13.40 | 19.89 |

e-MCLP is very suitable for PCCs with large waste volumes because vehicle capacity is filled faster when the volume of waste is large so that there are fewer pick-up points on one route. When there are fewer pick-up points in one route, the more routes there will be, and the development of this method is suitable for implementation. This model's savings in transportation costs will be felt when the number of routes increases because

vehicles will depart and return to SCC more often. That is, the closer CCP distance to the SCC is very beneficial for the vehicle. In this study, the selected location does not affect the investment cost because each candidate location requires a procurement cost of the same amount. However, the developed model can accommodate each candidate location requiring a different investment cost. Later, the selected location will provide a minimum total cost, including investment and transportation costs.

The best route was determined using the tabu search method to improve the results of the nearest neighbor. This is in line with the opinion found in [77,80,81] that metaheuristics are popular optimization problem-solving techniques to overcome the weaknesses of the heuristic method due to their ability to avoid being trapped in a local optimum solution [82]. The first route was found to be better than the original solution due to its ability to reduce the distance traveled by 15.6 km and the travel time by 0.33 h, thereby reducing the distance and travel time by 8%. Meanwhile, the optimal solution in the second route is the same as the initial solution, but the route completion sequence is reversed such that the first PCC visited using the nearest neighbor was the last in the tabu search method. This shows the nearest neighbor method also has the ability to provide the best solution, and this is in accordance with the findings of [41] that the nearest neighbor method produces the shortest route compared to other heuristic methods. The NN algorithm was able to minimize distribution costs [83] and could easily and quickly resolve problems for several small cities [84]. Furthermore, the initial solution was observed to be infeasible for the third route because the completion time, which was recorded to be 7.6 h, exceeds the planning time horizon, which is 7 h. The continuation of the iteration using the tabu search method changed the initially infeasible solution to feasible as indicated by the shortening of the completion time (CT) to 6.68 h with a total distance (D) of 225.5 km and a travel time (TT) of 5.01 h, saving 8.53% of travel time. This means the tabu search was able to reduce the distance and travel time by 15% as indicated by the 41.3 km and 0.92 h results when compared to the nearest neighbor method, as shown in Table 4.

The tabu search method was generally able to provide better performance than the nearest neighbor method. The metaheuristic approach gives better performance results than the heuristic approach [70]. The results showed the possibility of collecting all the smartphone waste in Yogyakarta Province using three routes. This can be completed in a day through the use of three vehicles or in three days through the use of one vehicle. The total distance required to be covered is 602.2 km with a travel time of 13.4 h and a total completion time of 19.89 h. This means the tabu search method generally saved 56.9 km (8.6%) distance and 1.25 h (8.5%) travel time.

Determination of smartphone waste collection routes in the province of Yogyakarta with one route picking up at several PCC points managed to save a mileage of 346.5 km compared to one route only picking up at one PCC point and a total of 30 pick-up points. If the smartphone waste collection is done once a week, this shorter distance can provide transportation cost savings of USD 2214.39 per year. The area of Yogyakarta Province is only 0.16% of the territory of Indonesia; if this model is implemented nationally, the estimated transportation cost savings will be more than USD 1 million.

This research is expected to be the initial framework in formulating e-waste management policies for the national formal channel. If this proposal is successfully implemented in Yogyakarta, it is likely to be implemented in other provinces in Indonesia. The developed model can also be used for other solid waste collection scenarios. The proposed e-MCLP model is very suitable for large e-waste because there is no need to proceed to route determination, considering that the supply from PCC may already meet vehicle capacity. One trip only picks up from a PCC and then returns to the SCC again. However, to use the proposed model, it is necessary to consider whether the community is willing to bring their large size/volume e-waste to the provided PCC. This research is also the first step in electronic waste management, which will then be followed by the next stage of management, which includes separation, repair, recycling, remanufacturing, or disposal.



## 5. Conclusions

Smartphone waste has a high economic value and has great potential. The tendency of people to store and dispose of smartphone waste is due to the absence of waste collection facilities and government regulations that specifically regulate electronic waste management mechanisms. With the public's willingness to bring smartphone waste to a collection point with a maximum reach of 11.2 km and the benefits that will be obtained, this is a challenge and an opportunity for the government to design an optimal collection channel. The design of the collection channel involves consumers as suppliers of electronic waste, primary collection centers (PCCs), and Secondary Collection Centers (SCC). Due to the small area of Yogyakarta Province, 1 SCC is sufficient to accommodate the supply of smartphone waste from all selected PCCs. Based on the results of calculations using e-MCLP, as many as 30 PCCs should be built, with a distribution of 1 PCC in Yogyakarta City, 13 PCCs in Gunung Kidul Regency, 6 PCCs in Bantul Regency, 6 PCCs in Sleman Regency, and 4 PCCs in Kulon Progo Regency. e-MCLP can produce the minimum number of primary collection facilities required to cover all consumers with the shortest distance from secondary collection facilities to minimize total costs, including investment and transportation costs, with a total cost of USD 3617.92 per year.

The best transportation route from PCC to SCC was determined using the nearest neighbor and tabu search method (NN-TB). The pick-up route starts and ends at SCC, and the result shows three routes to use in smartphone waste collection. These routes take three days to complete by using one vehicle or one day using three vehicles with a total time required of 19.89 h and a distance of 602.2 km.

Further research can expand the study of e-waste with a large volume because the large volume will affect the willingness of consumers to bring their e-waste and the need to calculate the capacity of the collection center. This research is expected to be the initial framework in formulating e-waste management policies for a formal national channel. Research can also be continued with the design of management following the collection of e-waste in a final disposal site, such as separation, repair, recycling, remanufacturing, or disposal.

**Author Contributions:** Conceptualization, D.P.S., N.A.M. and A.M.S.A.; methodology, D.P.S., N.A.M. and A.M.S.A.; software, D.P.S.; validation and analysis, D.P.S., N.A.M. and A.M.S.A.; data curation, D.P.S.; writing—original draft preparation, D.P.S.; writing—review and editing, N.A.M. and A.M.S.A.; visualization, D.P.S.; supervision, N.A.M. and A.M.S.A. All authors have read and agreed to the published version of the manuscript.

**Funding:** This paper is part of a PhD study that was financially supported by BUDI DN Grant from the Lembaga Pengelola Dana Penelitian (LPDP), Ministry of Finance, and Ministry of Education and Culture, Republic of Indonesia, grant number KET-193/LPDP.4/2020. The APC was funded by RTA 2021 grant number 3190/UN1/DITLIT/DIT-LIT/PT/2021 from Universitas Gadjah Mada.

**Institutional Review Board Statement:** Not applicable.

**Informed Consent Statement:** Not applicable.

**Data Availability Statement:** The data used to support the findings of this study are available from the corresponding author on request.

**Conflicts of Interest:** The authors declare no conflict of interest.

## Abbreviations

| | |
|---|---|
| B | Bantul Regency |
| CC | Collection Center |
| CT | Completion Time |
| D | Total Distance |
| e-MCLP | Extended Maximal Covering Location Problem |
| e-waste | Electronic Waste |

G　　　　　Gunung Kidul Regency
K　　　　　Kulon Progo Regency
LSCP　　　Location Set Covering Problem
MCLP　　　Maximal Covering Location Problem
NN　　　　Nearest Neighbor
NN-TS　　　Nearest Neighbor and Tabu Search
PCC　　　　Primary Collection Center
S　　　　　Sleman Regency
SCC　　　　Secondary Collection Center
TPA　　　　Tempat Pembuangan Akhir (Final Disposal Site)
TS　　　　　Tabu Search
TT　　　　　Travel Time
VRP　　　　Vehicle Routing Problem
Y　　　　　Yogyakarta City

## Appendix A

**Table A1.** Regencies/Cities and Districts in Yogyakarta Province.

| No | City/Regency | District | Code |
|---|---|---|---|
| 1 | Yogyakarta City (Y) | Danurejan | Y1 |
| | | Gedongtengen | Y2 |
| | | Gondokusuman | Y3 |
| | | Gondomanan | Y4 |
| | | Jetis | Y5 |
| | | Kotagede | Y6 |
| | | Kraton | Y7 |
| | | Mantrijeron | Y8 |
| | | Mergangsan | Y9 |
| | | Ngampilan | Y10 |
| | | Pakualaman | Y11 |
| | | Tegalrejo | Y12 |
| | | Umbulharjo | Y13 |
| | | Wirobrajan | Y14 |
| 2 | Gunung Kidul Regency (G) | Panggang | G1 |
| | | Purwosari | G2 |
| | | Paliyan | G3 |
| | | Saptosari | G4 |
| | | Tepus | G5 |
| | | Tanjungsari | G6 |
| | | Rongkop | G7 |
| | | Girisubo | G8 |
| | | Semanu | G9 |
| | | Ponjong | G10 |
| | | Karangmojo | G11 |
| | | Wonosari | G12 |
| | | Playen | G13 |
| | | Patuk | G14 |
| | | Gedangsari | G15 |
| | | Nglipar | G16 |
| | | Ngawen | G17 |
| | | Semin | G18 |

**Table A1.** *Cont.*

| No | City/Regency | District | Code |
|----|----|----|----|
| 3 | Bantul Regency (B) | Bambanglipuro | B1 |
| | | Banguntapan | B2 |
| | | Bantul | B3 |
| | | Dlingo | B4 |
| | | Imogiri | B5 |
| | | Jetis | B6 |
| | | Kasihan | B7 |
| | | Kretek | B8 |
| | | Pajangan | B9 |
| | | Pandak | B10 |
| | | Piyungan | B11 |
| | | Pleret | B12 |
| | | Pundong | B13 |
| | | Sanden | B14 |
| | | Sedayu | B15 |
| | | Sewon | B16 |
| | | Srandakan | B17 |
| 4 | Sleman Regency (S) | Moyudan | S1 |
| | | Godean | S2 |
| | | Minggir | S3 |
| | | Gamping | S4 |
| | | Seyegan | S5 |
| | | Sleman | S6 |
| | | Ngaglik | S7 |
| | | Mlati | S8 |
| | | Tempel | S9 |
| | | Turi | S10 |
| | | Prambanan | S11 |
| | | Kalasan | S12 |
| | | Berbah | S13 |
| | | Ngemplak | S14 |
| | | Pakem | S15 |
| | | Depok | S16 |
| | | Cangkringan | S17 |
| 5 | Kulon Progo Regency (K) | Temon | K1 |
| | | Wates | K2 |
| | | Panjatan | K3 |
| | | Galur | K4 |
| | | Lendah | K5 |
| | | Sentolo | K6 |
| | | Pengasih | K7 |
| | | Kokap | K8 |
| | | Girimulyo | K9 |
| | | Nanggulan | K10 |
| | | Samigaluh | K11 |
| | | Kalibawang | K12 |

## Appendix B

**Table A2.** The distance matrice between PCC candidates and from the PCC candidates to the SCC in Yogyakarta City.

| Distances (km) | Y1 | Y2 | Y3 | Y4 | Y5 | Y6 | Y7 | Y8 | Y9 | Y10 | Y11 | Y12 | Y13 | Y14 |
|----|----|----|----|----|----|----|----|----|----|----|----|----|----|----|
| Y1 | 0 | 3 | 1.9 | 2.7 | 2.8 | 6.2 | 4.1 | 5.3 | 4.4 | 3.2 | 1.9 | 5.1 | 4.5 | 4.3 |
| Y2 | 3 | 0 | 2.7 | 2.2 | 2.1 | 6.9 | 3 | 4 | 5 | 2.1 | 2.3 | 2.9 | 6 | 2.5 |
| Y3 | 1.9 | 2.7 | 0 | 4.6 | 3 | 5.5 | 6.4 | 7.3 | 5 | 5.5 | 3.8 | 4.7 | 4.1 | 6.5 |
| Y4 | 2.7 | 2.2 | 4.6 | 0 | 3.5 | 5.1 | 2.4 | 3.6 | 3.2 | 1.6 | 1.1 | 4.7 | 3.3 | 2.6 |

**Table A2.** *Cont.*

| Distances (km) | Y1 | Y2 | Y3 | Y4 | Y5 | Y6 | Y7 | Y8 | Y9 | Y10 | Y11 | Y12 | Y13 | Y14 |
|---|---|---|---|---|---|---|---|---|---|---|---|---|---|---|
| Y5 | 2.8 | 2.1 | 3 | 3.5 | 0 | 7.9 | 5.1 | 4.9 | 6.6 | 3 | 4 | 1.8 | 6.3 | 3.3 |
| Y6 | 6.2 | 6.9 | 5.5 | 5.1 | 7.9 | 0 | 5.2 | 5.2 | 2.8 | 5.8 | 4.3 | 9.5 | 2.1 | 6.6 |
| Y7 | 4.1 | 3 | 6.4 | 2.4 | 5.1 | 5.2 | 0 | 2.4 | 3.5 | 2 | 2.9 | 5.1 | 4.3 | 2.2 |
| Y8 | 5.3 | 4 | 7.3 | 3.6 | 4.9 | 5.2 | 2.4 | 0 | 3.5 | 2.7 | 4 | 5.9 | 4.3 | 2.8 |
| Y9 | 4.4 | 5 | 5 | 3.2 | 6.6 | 2.8 | 3.5 | 3.5 | 0 | 4.1 | 2.9 | 7.6 | 2 | 4.9 |
| Y10 | 3.2 | 2.1 | 5.5 | 1.6 | 3 | 5.8 | 2 | 2.7 | 4.1 | 0 | 2 | 3.9 | 4.2 | 1.3 |
| Y11 | 1.9 | 2.3 | 3.8 | 1.1 | 4 | 4.3 | 2.9 | 4 | 2.9 | 2 | 0 | 5 | 2.7 | 3.1 |
| Y12 | 5.1 | 2.9 | 4.7 | 4.7 | 1.8 | 9.5 | 5.1 | 5.9 | 7.6 | 3.9 | 5 | 0 | 7.3 | 3.2 |
| Y13 | 4.5 | 6 | 4.1 | 3.3 | 6.3 | 2.1 | 4.3 | 4.3 | 2 | 4.2 | 2.7 | 7.3 | 0 | 5.7 |
| Y14 | 4.3 | 2.5 | 6.5 | 2.6 | 3.3 | 6.6 | 2.2 | 2.8 | 4.9 | 1.3 | 3.1 | 3.2 | 5.7 | 0 |
| SCC | 14.4 | 15.5 | 16.2 | 13.1 | 17.2 | 9 | 14 | 13.1 | 11 | 14.8 | 12.5 | 18.6 | 10.6 | 15.5 |

**Table A3.** The distance matrice between PCC candidates and from the PCC candidates to the SCC in Gunung Kidul Regency.

| Distances (km) | G1 | G2 | G3 | G4 | G5 | G6 | G7 | G8 | G9 | G10 | G11 | G12 | G13 | G14 | G15 | G16 | G17 | G18 |
|---|---|---|---|---|---|---|---|---|---|---|---|---|---|---|---|---|---|---|
| G1 | 0 | 11.1 | 19 | 12.7 | 35.7 | 23.9 | 55.9 | 51.2 | 34.2 | 42.3 | 38.5 | 27.9 | 28.8 | 35.7 | 49.4 | 40.3 | 50.4 | 51.6 |
| G2 | 11.1 | 0 | 29 | 22.7 | 45.7 | 33.9 | 65.9 | 61.2 | 44.3 | 52.3 | 48.5 | 37.8 | 36.5 | 45.7 | 59.4 | 50.3 | 60.4 | 61.7 |
| G3 | 19 | 29 | 0 | 11.8 | 28.4 | 17.5 | 39.5 | 41.6 | 17.8 | 26.8 | 22 | 11.4 | 10 | 19.3 | 32.9 | 23.9 | 33.9 | 35.2 |
| G4 | 12.7 | 22.7 | 11.8 | 0 | 28.5 | 16.7 | 48.7 | 44 | 27.1 | 36.1 | 31.3 | 20.7 | 19.3 | 28.5 | 42.2 | 33.1 | 43.2 | 44.5 |
| G5 | 35.7 | 45.7 | 28.4 | 28.5 | 0 | 15.2 | 22.7 | 16.3 | 16.8 | 25.5 | 25.3 | 23.4 | 31.3 | 38.2 | 51.9 | 34.7 | 37.2 | 37.2 |
| G6 | 23.9 | 33.9 | 17.5 | 16.7 | 15.2 | 0 | 27.6 | 30.7 | 20.4 | 29.2 | 27.7 | 17.8 | 25.7 | 32.6 | 46.3 | 29.1 | 40.9 | 40.9 |
| G7 | 55.9 | 65.9 | 39.5 | 48.7 | 22.7 | 27.6 | 0 | 15.9 | 21.7 | 18.8 | 26.2 | 29.2 | 38.3 | 45.2 | 53 | 41 | 38.1 | 38.1 |
| G8 | 51.2 | 61.2 | 41.6 | 44 | 16.3 | 30.7 | 15.9 | 0 | 27.2 | 30.9 | 35.7 | 34.5 | 43.6 | 50.2 | 62.5 | 46.3 | 47.6 | 47.6 |
| G9 | 34.2 | 44.3 | 17.8 | 27.1 | 16.8 | 20.4 | 21.7 | 27.2 | 0 | 9.1 | 8.5 | 7.5 | 16.6 | 23.5 | 35.4 | 19.3 | 20.4 | 20.1 |
| G10 | 42.3 | 52.3 | 26.8 | 36.1 | 25.5 | 29.2 | 18.8 | 30.9 | 9.1 | 0 | 8.4 | 14.4 | 22.5 | 29.4 | 35.3 | 21.6 | 20.3 | 20.3 |
| G11 | 38.5 | 48.5 | 22 | 31.3 | 25.3 | 27.7 | 26.2 | 35.7 | 8.5 | 8.4 | 0 | 10.6 | 18.7 | 22.7 | 28.6 | 13.3 | 13.7 | 13.7 |
| G12 | 27.9 | 37.8 | 11.4 | 20.7 | 23.4 | 17.8 | 29.2 | 34.5 | 7.5 | 14.4 | 10.6 | 0 | 8.5 | 15.4 | 29.1 | 11.7 | 23.8 | 23.8 |
| G13 | 28.8 | 36.5 | 10 | 19.3 | 31.3 | 25.7 | 38.3 | 43.6 | 16.6 | 22.5 | 18.7 | 8.5 | 0 | 11.1 | 24.8 | 15.7 | 25.7 | 31.9 |
| G14 | 35.7 | 45.7 | 19.3 | 28.5 | 38.2 | 32.6 | 45.2 | 50.2 | 23.5 | 29.4 | 22.7 | 15.4 | 11.1 | 0 | 19.6 | 14.6 | 24.5 | 31.4 |
| G15 | 49.4 | 59.4 | 32.9 | 42.2 | 51.9 | 46.3 | 53 | 62.5 | 35.4 | 35.3 | 28.6 | 29.1 | 24.8 | 19.6 | 0 | 22 | 16.6 | 24.3 |
| G16 | 40.3 | 50.3 | 23.9 | 33.1 | 34.7 | 29.1 | 41 | 46.3 | 19.3 | 21.6 | 13.3 | 11.7 | 15.7 | 14.6 | 22 | 0 | 10.6 | 20.6 |
| G17 | 50.4 | 60.4 | 33.9 | 43.2 | 37.2 | 40.9 | 38.1 | 47.6 | 20.4 | 20.3 | 13.7 | 23.8 | 25.7 | 24.5 | 16.6 | 10.6 | 0 | 8.6 |
| G18 | 51.6 | 61.7 | 35.2 | 44.5 | 37.2 | 40.9 | 38.1 | 47.6 | 20.1 | 20.3 | 13.7 | 23.8 | 31.9 | 31.4 | 24.3 | 20.6 | 8.6 | 0 |

**Table A4.** The distance matrice between PCC candidates and from the PCC candidates to the SCC in Bantul Regency.

| Distances (km) | B1 | B2 | B3 | B4 | B5 | B6 | B7 | B8 | B9 | B10 | B11 | B12 | B13 | B14 | B15 | B16 | B17 |
|---|---|---|---|---|---|---|---|---|---|---|---|---|---|---|---|---|---|
| B1 | 0 | 25.7 | 12 | 24.9 | 12.7 | 12.5 | 17.2 | 6.5 | 13.7 | 5.8 | 31.5 | 21.4 | 4.5 | 8.1 | 22.6 | 15.8 | 10.4 |
| B2 | 25.7 | 0 | 16.2 | 22.5 | 14.3 | 12.8 | 14.1 | 28.3 | 18.9 | 23.3 | 8.4 | 9.8 | 23.8 | 28.8 | 26.4 | 7.5 | 30.5 |
| B3 | 12 | 16.2 | 0 | 19.3 | 7.1 | 4.4 | 10.2 | 14 | 7.3 | 8.3 | 23.1 | 13.4 | 10.8 | 13.4 | 18.9 | 7.1 | 15.4 |
| B4 | 24.9 | 22.5 | 19.3 | 0 | 13.6 | 16.9 | 30 | 29.6 | 28.2 | 26.5 | 13.7 | 12.7 | 25.1 | 31.9 | 37.2 | 24.3 | 33.7 |
| B5 | 12.7 | 14.3 | 7.1 | 13.6 | 0 | 3.5 | 16.7 | 16.2 | 14.9 | 13.2 | 18.7 | 9.6 | 11.8 | 18.5 | 23.8 | 11.2 | 20.3 |
| B6 | 12.5 | 12.8 | 4.4 | 16.9 | 3.5 | 0 | 13.3 | 16 | 11.5 | 12.4 | 18.6 | 9 | 11.6 | 18.3 | 23.1 | 7.6 | 19.6 |
| B7 | 17.2 | 14.1 | 10.2 | 30 | 16.7 | 13.3 | 0 | 22.1 | 6.6 | 14.8 | 20.8 | 16.8 | 20.3 | 19.3 | 15.1 | 6.5 | 20.9 |
| B8 | 6.5 | 28.3 | 14 | 29.6 | 16.2 | 16 | 22.1 | 0 | 18.7 | 10.7 | 35 | 24.9 | 6.7 | 6.5 | 27.5 | 19.3 | 11 |
| B9 | 13.7 | 18.9 | 7.3 | 28.2 | 14.9 | 11.5 | 6.6 | 18.7 | 0 | 9.2 | 27 | 17.6 | 17.3 | 13.8 | 13.2 | 11.1 | 15.4 |

**Table A4.** *Cont.*

| Distances (km) | B1 | B2 | B3 | B4 | B5 | B6 | B7 | B8 | B9 | B10 | B11 | B12 | B13 | B14 | B15 | B16 | B17 |
|---|---|---|---|---|---|---|---|---|---|---|---|---|---|---|---|---|---|
| B10 | 5.8 | 23.3 | 8.3 | 26.5 | 13.2 | 12.4 | 14.8 | 10.7 | 9.2 | 0 | 30 | 20.9 | 9.7 | 7.8 | 18.2 | 14.3 | 5.5 |
| B11 | 31.5 | 8.4 | 23.1 | 13.7 | 18.7 | 18.6 | 20.8 | 35 | 27 | 30 | 0 | 9.6 | 30 | 35.5 | 33 | 14.2 | 37.1 |
| B12 | 21.4 | 9.8 | 13.4 | 12.7 | 9.6 | 9 | 16.8 | 24.9 | 17.6 | 20.9 | 9.6 | 0 | 20.5 | 27.3 | 28.7 | 10.2 | 29 |
| B13 | 4.5 | 23.8 | 10.8 | 25.1 | 11.8 | 11.6 | 20.3 | 6.7 | 17.3 | 9.7 | 30 | 20.5 | 0 | 10.3 | 25.8 | 14.9 | 11.8 |
| B14 | 8.1 | 28.8 | 13.4 | 31.9 | 18.5 | 18.3 | 19.3 | 6.5 | 13.8 | 7.8 | 35.5 | 27.3 | 10.3 | 0 | 22.1 | 19.9 | 5.3 |
| B15 | 22.6 | 26.4 | 18.9 | 37.2 | 23.8 | 23.1 | 15.1 | 27.5 | 13.2 | 18.2 | 33 | 28.7 | 25.8 | 22.1 | 0 | 18.8 | 24.1 |
| B16 | 15.8 | 7.5 | 7.1 | 24.3 | 11.2 | 7.6 | 6.5 | 19.3 | 11.1 | 14.3 | 14.2 | 10.2 | 14.9 | 19.9 | 18.8 | 0 | 21.4 |
| B17 | 10.4 | 30.5 | 15.4 | 33.7 | 20.3 | 19.6 | 20.9 | 11 | 15.4 | 5.5 | 37.1 | 29 | 11.8 | 5.3 | 24.1 | 21.4 | 0 |
| SCC | 23.6 | 9.2 | 15.6 | 15.8 | 11.6 | 11.1 | 16.8 | 27.1 | 17.7 | 23.5 | 10.1 | 4.8 | 22.6 | 29.4 | 29.1 | 10.6 | 30.7 |

**Table A5.** The distance matrice between PCC candidates and from the PCC candidates to the SCC in Sleman Regency.

| Distances (km) | S1 | S2 | S3 | S4 | S5 | S6 | S7 | S8 | S9 | S10 | S11 | S12 | S13 | S14 | S15 | S16 | S17 |
|---|---|---|---|---|---|---|---|---|---|---|---|---|---|---|---|---|---|
| S1 | 0 | 6.0 | 6.9 | 10.6 | 11.6 | 19.3 | 23.8 | 16.2 | 16.5 | 26.8 | 38.4 | 31.1 | 29.9 | 31.5 | 30.5 | 22.6 | 37.8 |
| S2 | 6.0 | 0 | 8.5 | 6.3 | 6.6 | 13.1 | 17.8 | 8.3 | 14.3 | 21.9 | 32.4 | 25.7 | 25.6 | 25.5 | 24.6 | 16.6 | 31.9 |
| S3 | 6.9 | 8.5 | 0 | 14.9 | 8.4 | 16.1 | 21.9 | 13 | 13.3 | 23.6 | 38.7 | 30.9 | 32.3 | 30.4 | 27.3 | 0.92 | 34.6 |
| S4 | 10.6 | 6.3 | 14.9 | 0 | 12.9 | 15.8 | 16.2 | 9.3 | 19.9 | 25 | 30.9 | 23.6 | 19.3 | 24 | 25 | 15.1 | 29.8 |
| S5 | 11.6 | 6.6 | 8.4 | 12.9 | 0 | 8.6 | 14.1 | 5.2 | 8.8 | 19.4 | 31.5 | 23.6 | 24.5 | 22.6 | 20.9 | 15.1 | 27.5 |
| S6 | 19.3 | 13.1 | 16.1 | 15.8 | 8.6 | 0 | 8.2 | 2.7 | 7.5 | 9.4 | 29.3 | 18.9 | 22.9 | 16 | 12.1 | 13.5 | 19.4 |
| S7 | 23.8 | 17.8 | 21.9 | 16.2 | 14.1 | 8.2 | 0 | 9.3 | 14.5 | 13.9 | 23.1 | 12.8 | 16.7 | 8.9 | 11.8 | 7.7 | 14.8 |
| S8 | 16.2 | 8.3 | 13.0 | 9.3 | 5.2 | 2.7 | 9.3 | 0 | 11.5 | 15.3 | 26.3 | 19 | 18.9 | 19.4 | 18 | 10.5 | 25.3 |
| S9 | 16.5 | 14.3 | 13.3 | 19.9 | 8.8 | 7.5 | 14.5 | 11.5 | 0 | 11.4 | 35 | 24.1 | 27.6 | 23.2 | 15.2 | 19.3 | 22.5 |
| S10 | 26.8 | 21.9 | 23.6 | 25 | 19.4 | 9.4 | 13.9 | 15.3 | 11.4 | 0 | 32.4 | 20.8 | 29.8 | 16.8 | 10 | 21.6 | 12.9 |
| S11 | 38.4 | 32.4 | 38.7 | 30.9 | 31.5 | 29.3 | 23.1 | 26.3 | 35 | 32.4 | 0 | 11.8 | 10.6 | 18.9 | 28.2 | 15.7 | 25.9 |
| S12 | 31.1 | 25.7 | 30.9 | 23.6 | 23.6 | 18.9 | 12.8 | 19 | 24.1 | 20.8 | 11.8 | 0 | 9 | 7.2 | 16.4 | 12.2 | 14.3 |
| S13 | 29.9 | 25.6 | 32.3 | 19.3 | 24.5 | 22.9 | 16.7 | 18.9 | 27.6 | 29.8 | 10.6 | 9 | 0 | 16.2 | 26.9 | 8.8 | 23.2 |
| S14 | 31.5 | 25.5 | 30.4 | 24 | 22.6 | 16 | 8.9 | 19.4 | 23.2 | 16.8 | 18.9 | 7.2 | 16.2 | 0 | 11.9 | 11.4 | 10.2 |
| S15 | 30.5 | 24.6 | 27.3 | 25 | 20.9 | 12.1 | 11.8 | 18 | 15.2 | 10 | 28.2 | 16.4 | 26.9 | 11.9 | 0 | 16.7 | 7.8 |
| S16 | 22.6 | 16.6 | 0.92 | 15.1 | 15.1 | 13.5 | 7.7 | 10.5 | 19.3 | 21.6 | 15.7 | 12.2 | 8.8 | 11.4 | 16.7 | 0 | 21.4 |
| S17 | 37.8 | 31.9 | 34.6 | 29.8 | 27.5 | 19.4 | 14.8 | 25.3 | 22.5 | 12.9 | 25.9 | 14.3 | 23.2 | 10.2 | 7.8 | 21.4 | 0 |
| SCC | 29.8 | 25.5 | 34.0 | 19.1 | 30.7 | 29.8 | 25.8 | 27.1 | 33.8 | 38.7 | 17.7 | 17.8 | 9.1 | 23.9 | 34.6 | 20.1 | 30.9 |

**Table A6.** The distance matrice between PCC candidates and from the PCC candidates to the SCC in Kulon Progo Regency.

| Distances (km) | K1 | K2 | K3 | K4 | K5 | K6 | K7 | K8 | K9 | K10 | K11 | K12 |
|---|---|---|---|---|---|---|---|---|---|---|---|---|
| K1 | 0.0 | 7.6 | 11.3 | 19 | 16.8 | 19.7 | 15.8 | 9.1 | 27.6 | 23.1 | 47.9 | 41.9 |
| K2 | 7.6 | 0 | 3.7 | 12.1 | 9.2 | 13.4 | 11.4 | 13.4 | 29.1 | 18.3 | 41.6 | 35.6 |
| K3 | 11.3 | 3.7 | 0 | 10.9 | 8.8 | 15.4 | 11.6 | 17.1 | 31.1 | 20.3 | 43.5 | 37.5 |
| K4 | 19.0 | 12.1 | 10.9 | 0 | 6 | 13.6 | 19.8 | 24.8 | 29.3 | 22.6 | 42 | 35.8 |
| K5 | 16.8 | 9.2 | 8.8 | 6 | 0 | 8.5 | 14.1 | 19.2 | 24.2 | 17.5 | 36.7 | 30.7 |
| K6 | 19.7 | 13.4 | 15.4 | 13.6 | 8.5 | 0 | 9.5 | 17 | 19.1 | 12.4 | 31.5 | 25.5 |
| K7 | 15.8 | 11.4 | 11.6 | 19.8 | 14.1 | 9.5 | 0 | 9.4 | 13.7 | 10.8 | 32.4 | 26.4 |
| K8 | 9.1 | 13.4 | 17.1 | 24.8 | 19.2 | 17 | 9.4 | 0 | 23 | 19.2 | 37.9 | 31.8 |
| K9 | 27.6 | 29.1 | 31.1 | 29.3 | 24.2 | 19.1 | 13.7 | 23 | 0 | 6.4 | 20.5 | 14.5 |
| K10 | 23.1 | 18.3 | 20.3 | 22.6 | 17.5 | 12.4 | 10.8 | 19.2 | 6.4 | 0 | 21.3 | 17.7 |
| K11 | 47.9 | 41.6 | 43.5 | 42 | 36.7 | 31.5 | 32.4 | 37.9 | 20.5 | 21.3 | 0 | 8.5 |
| K12 | 41.9 | 35.6 | 37.5 | 35.8 | 30.7 | 25.5 | 26.4 | 31.8 | 14.5 | 17.7 | 8.5 | 0 |
| SCC | 48.8 | 42.2 | 41.8 | 33.8 | 35.5 | 35.8 | 40.8 | 49.2 | 42.2 | 35.5 | 46.6 | 40.6 |

## Appendix C

**Table A7.** The selected PCC numbers and locations.

| City/Regency | PCC Number | Location |
|---|---|---|
| Yogyakarta City | 1 | Kotagede |
| Gunung Kidul Regency | 2 | Panggang |
| | 3 | Purwosari |
| | 4 | Saptosari |
| | 5 | Tepus |
| | 6 | Tanjungsari |
| | 7 | Rongkop |
| | 8 | Girisubo |
| | 9 | Playen |
| | 10 | Patuk |
| | 11 | Gedangsari |
| | 12 | Ngglipar |
| | 13 | Ngawen |
| | 14 | Semanu |
| Bantul Regency | 15 | Dlingo |
| | 16 | Pandak |
| | 17 | Pleret |
| | 18 | Sedayu |
| | 19 | Sewon |
| | 20 | Pundon |
| Sleman Regency | 21 | Godean |
| | 22 | Sleman |
| | 23 | Prambanan |
| | 24 | Kalasan |
| | 25 | Cangkringan |
| | 26 | Berbah |
| Kulon Progo Regency | 27 | Lendah |
| | 28 | Kokap |
| | 29 | Nanggulan |
| | 30 | Kalibawang |

## Appendix D

**Table A8.** The distance matrice between the selected PCCs and from the selected PCC to the SCC.

| Distances (km) | | SCC | PCC | | | | | | | | | | | | | | |
|---|---|---|---|---|---|---|---|---|---|---|---|---|---|---|---|---|---|
| | | | 1 | 2 | 3 | 4 | 5 | 6 | 7 | 8 | 9 | 10 | 11 | 12 | 13 | 14 | 15 |
| SCC | | 0 | 9 | 30.6 | 31 | 42.6 | 54.6 | 47.4 | 61.6 | 67 | 39.9 | 25.6 | 23.3 | 38.4 | 33.4 | 43.2 | 22.6 |
| PCC | 1 | 9 | 0 | 31.2 | 31.7 | 43.9 | 57.4 | 50.2 | 64.3 | 69.8 | 42.7 | 30.2 | 22.7 | 38.7 | 33.8 | 43.7 | 22.9 |
| | 2 | 30.6 | 31.2 | 0 | 11.1 | 12.7 | 35.7 | 23.9 | 55.9 | 51.2 | 34.2 | 28.8 | 35.7 | 49.4 | 40.3 | 50.4 | 22.5 |
| | 3 | 31 | 31.7 | 11.1 | 0 | 22.7 | 45.7 | 33.9 | 65.9 | 61.2 | 44.3 | 36.5 | 45.7 | 59.4 | 50.3 | 60.4 | 13.5 |
| | 4 | 42.6 | 43.9 | 12.7 | 22.7 | 0 | 28.5 | 16.7 | 48.7 | 44 | 27.1 | 19.3 | 28.5 | 42.2 | 33.1 | 43.2 | 35.2 |
| | 5 | 54.6 | 57.4 | 35.7 | 45.7 | 28.5 | 0 | 15.2 | 22.7 | 16.3 | 16.8 | 31.3 | 38.2 | 51.9 | 34.7 | 37.2 | 59 |
| | 6 | 47.4 | 50.2 | 23.9 | 33.9 | 16.7 | 15.2 | 0 | 27.6 | 30.7 | 20.4 | 25.7 | 32.6 | 46.3 | 29.1 | 40.9 | 46.4 |
| | 7 | 61.6 | 64.3 | 55.9 | 65.9 | 48.7 | 22.7 | 27.6 | 0 | 15.9 | 21.7 | 38.3 | 45.2 | 53 | 41 | 38.1 | 70.1 |
| | 8 | 67 | 69.8 | 51.2 | 61.2 | 44 | 16.3 | 30.7 | 15.9 | 0 | 27.2 | 43.6 | 50.2 | 62.5 | 46.3 | 47.6 | 72.3 |
| | 9 | 39.9 | 42.7 | 34.2 | 44.3 | 27.1 | 16.8 | 20.4 | 21.7 | 27.2 | 0 | 16.6 | 23.5 | 35.4 | 19.3 | 20.4 | 48.4 |
| | 10 | 25.6 | 30.2 | 28.8 | 36.5 | 19.3 | 31.3 | 25.7 | 38.3 | 43.6 | 16.6 | 0 | 11.1 | 24.8 | 15.7 | 25.7 | 36.7 |
| | 11 | 23.3 | 22.7 | 35.7 | 45.7 | 28.5 | 38.2 | 32.6 | 45.2 | 50.2 | 23.5 | 11.1 | 0 | 19.6 | 14.6 | 24.5 | 45.9 |
| | 12 | 38.4 | 38.7 | 49.4 | 59.4 | 42.2 | 51.9 | 46.3 | 53 | 62.5 | 35.4 | 24.8 | 19.6 | 0 | 22 | 16.6 | 59.6 |

**Table A8.** *Cont.*

| Distances (km) | | SCC | PCC | | | | | | | | | | | | | | |
|---|---|---|---|---|---|---|---|---|---|---|---|---|---|---|---|---|---|
| | | | 1 | 2 | 3 | 4 | 5 | 6 | 7 | 8 | 9 | 10 | 11 | 12 | 13 | 14 | 15 |
| PCC | 13 | 33.4 | 33.8 | 40.3 | 50.3 | 33.1 | 34.7 | 29.1 | 41 | 46.3 | 19.3 | 15.7 | 14.6 | 22 | 0 | 10.6 | 50.5 |
| | 14 | 43.2 | 43.7 | 50.4 | 60.4 | 43.2 | 37.2 | 40.9 | 38.1 | 47.6 | 20.4 | 25.7 | 24.5 | 16.6 | 10.6 | 0 | 60.6 |
| | 15 | 22.6 | 22.9 | 22.5 | 13.5 | 35.2 | 59 | 46.4 | 70.1 | 72.3 | 48.4 | 36.7 | 45.9 | 59.6 | 50.5 | 60.6 | 0 |
| | 16 | 15.8 | 22.7 | 35.7 | 31.5 | 28.6 | 40.6 | 35 | 47.5 | 52.9 | 25.9 | 11.6 | 20.6 | 34.3 | 25.1 | 35.2 | 25.1 |
| | 17 | 23.5 | 21.8 | 30.7 | 25.7 | 43.2 | 65.4 | 54.4 | 72.4 | 77.7 | 50.7 | 36.4 | 43.5 | 61 | 50 | 60.1 | 9.7 |
| | 18 | 4.8 | 10 | 28 | 28.4 | 39.5 | 51.6 | 45.9 | 58.5 | 63.8 | 36.9 | 22.5 | 20.6 | 36.7 | 31.7 | 41.5 | 20.5 |
| | 19 | 29.1 | 24.9 | 41.3 | 40.4 | 53.9 | 76.1 | 65.1 | 83.1 | 88.4 | 61.4 | 47.1 | 45.6 | 61.7 | 56.7 | 66.6 | 25.8 |
| | 20 | 10.6 | 8.5 | 28.8 | 32.7 | 41.4 | 61.5 | 55.9 | 68.4 | 73.8 | 46.8 | 34.4 | 26.8 | 45.3 | 37.9 | 47.8 | 14.9 |
| | 21 | 25.50 | 16.2 | 44 | 48 | 56.6 | 76.7 | 71.1 | 83.7 | 89 | 62 | 49.6 | 42 | 53.2 | 53.1 | 63 | 32.1 |
| | 22 | 29.80 | 22.4 | 52.2 | 56.2 | 67.3 | 77 | 71.3 | 83.9 | 89.3 | 62.3 | 49.8 | 40.3 | 51.3 | 53.4 | 59.3 | 26.4 |
| | 23 | 17.70 | 18.10 | 46.10 | 46.5 | 45.8 | 55.5 | 49.9 | 62.5 | 67.8 | 40.8 | 28.4 | 13.8 | 24.8 | 31.9 | 41.8 | 40 |
| | 24 | 17.80 | 15.3 | 46.2 | 46.6 | 50.8 | 60.5 | 54.8 | 67.4 | 72.8 | 45.8 | 33.3 | 22.1 | 30.2 | 36.9 | 38.6 | 32.9 |
| | 25 | 9.10 | 7.5 | 35.3 | 35.7 | 45 | 54.7 | 49.1 | 61.7 | 67 | 40 | 27.6 | 20 | 33.4 | 31.1 | 41 | 27 |
| | 26 | 30.90 | 28.8 | 59.6 | 60.1 | 63.2 | 72.9 | 67.2 | 79.8 | 85.2 | 58.2 | 45.7 | 34.5 | 37.1 | 49.3 | 45.6 | 45 |
| | 27 | 35.50 | 33.8 | 42.7 | 35 | 55.2 | 77.4 | 66.4 | 84.4 | 89.7 | 62.7 | 48.4 | 54.6 | 68.3 | 62 | 72.1 | 20.6 |
| | 28 | 49.20 | 40.9 | 61.5 | 53.8 | 74 | 96.2 | 85.2 | 103.2 | 108.5 | 81.5 | 67.2 | 65.7 | 81.9 | 76.8 | 86.7 | 24 |
| | 29 | 35.50 | 31.2 | 50.1 | 51.6 | 62.7 | 86.7 | 73.9 | 93.6 | 98.9 | 72 | 59.5 | 52 | 68.1 | 63.1 | 73 | 10.3 |
| | 30 | 40.60 | 36.4 | 60 | 59 | 72.5 | 91.8 | 83.8 | 98.8 | 104.1 | 77.1 | 64.7 | 57.1 | 65.7 | 68.2 | 74.1 | 44.4 |

**Table A9.** The distance matrice between the selected PCCs and from the selected PCC to the SCC.

| Distances (km) | | PCC | | | | | | | | | | | | | | |
|---|---|---|---|---|---|---|---|---|---|---|---|---|---|---|---|---|
| | | 16 | 17 | 18 | 19 | 20 | 21 | 22 | 23 | 24 | 25 | 26 | 27 | 28 | 29 | 30 |
| SCC | | 15.8 | 23.5 | 4.8 | 29.1 | 10.6 | 25.5 | 29.8 | 17.7 | 17.8 | 9.1 | 30.9 | 35.5 | 49.2 | 35.5 | 40.6 |
| PCC | 1 | 22.7 | 21.8 | 10 | 24.9 | 8.5 | 16.2 | 22.4 | 18.1 | 15.3 | 7.5 | 28.8 | 33.8 | 40.9 | 31.2 | 36.4 |
| | 2 | 35.7 | 30.7 | 28 | 41.3 | 28.8 | 44 | 52.2 | 46.1 | 46.2 | 35.3 | 59.6 | 42.7 | 61.5 | 50.1 | 60 |
| | 3 | 31.5 | 25.7 | 28.4 | 40.4 | 32.7 | 48 | 56.2 | 46.5 | 46.6 | 35.7 | 60.1 | 35 | 53.8 | 51.6 | 59 |
| | 4 | 28.6 | 43.2 | 39.5 | 53.9 | 41.4 | 56.6 | 67.3 | 45.8 | 50.8 | 45 | 63.2 | 55.2 | 74 | 62.7 | 72.5 |
| | 5 | 40.6 | 65.4 | 51.6 | 76.1 | 61.5 | 76.7 | 77 | 55.5 | 60.5 | 54.7 | 72.9 | 77.4 | 96.2 | 86.7 | 91.8 |
| | 6 | 35 | 54.4 | 45.9 | 65.1 | 55.9 | 71.1 | 71.3 | 49.9 | 54.8 | 49.1 | 67.2 | 66.4 | 85.2 | 73.9 | 83.8 |
| | 7 | 47.5 | 72.4 | 58.5 | 83.1 | 68.4 | 83.7 | 83.9 | 62.5 | 67.4 | 61.7 | 79.8 | 84.4 | 103.2 | 93.6 | 98.8 |
| | 8 | 52.9 | 77.7 | 63.8 | 88.4 | 73.8 | 89 | 89.3 | 67.8 | 72.8 | 67 | 85.2 | 89.7 | 108.5 | 98.9 | 104.1 |
| | 9 | 25.9 | 50.7 | 36.9 | 61.4 | 46.8 | 62 | 62.3 | 40.8 | 45.8 | 40 | 58.2 | 62.7 | 81.5 | 72 | 77.1 |
| | 10 | 11.6 | 36.4 | 22.5 | 47.1 | 34.4 | 49.6 | 49.8 | 28.4 | 33.3 | 27.6 | 45.7 | 48.4 | 67.2 | 59.5 | 64.7 |
| | 11 | 20.6 | 43.5 | 20.6 | 45.6 | 26.8 | 42 | 40.3 | 13.8 | 22.1 | 20 | 34.5 | 54.6 | 65.7 | 52 | 57.1 |
| | 12 | 34.3 | 61 | 36.7 | 61.7 | 45.3 | 53.2 | 51.3 | 24.8 | 30.2 | 33.4 | 37.1 | 68.3 | 81.9 | 68.1 | 65.7 |
| | 13 | 25.1 | 50 | 31.7 | 56.7 | 37.9 | 53.1 | 53.4 | 31.9 | 36.9 | 31.1 | 49.3 | 62 | 76.8 | 63.1 | 68.2 |
| | 14 | 35.2 | 60.1 | 41.5 | 66.6 | 47.8 | 63 | 59.3 | 41.8 | 38.6 | 41 | 45.6 | 72.1 | 86.7 | 73 | 74.1 |
| | 15 | 25.1 | 9.7 | 20.5 | 25.8 | 14.9 | 32.1 | 26.4 | 40 | 32.9 | 27 | 45 | 20.6 | 24 | 10.3 | 44.4 |
| | 16 | 0 | 26.5 | 12.7 | 37.2 | 24.3 | 37.8 | 46 | 25.1 | 30 | 21.1 | 42.4 | 38.5 | 57.3 | 46 | 55.8 |
| | 17 | 26.5 | 0 | 20.9 | 18.2 | 14.3 | 25.5 | 34.2 | 36.9 | 36.6 | 26.9 | 50.1 | 14.2 | 33 | 27 | 36.8 |
| | 18 | 12.7 | 20.9 | 0 | 28.7 | 10.2 | 25.1 | 33.3 | 19.6 | 19.7 | 11 | 32.9 | 33.9 | 52.7 | 35 | 40.2 |
| | 19 | 37.2 | 18.2 | 28.7 | 0 | 18.8 | 10.8 | 26.4 | 40 | 32.9 | 30 | 45 | 15.7 | 24 | 10.3 | 20.7 |
| | 20 | 24.3 | 14.3 | 10.2 | 18.8 | 0 | 17.9 | 26.1 | 23.9 | 23.6 | 13.9 | 37 | 26.4 | 41.6 | 27.9 | 33 |
| | 21 | 37.8 | 25.5 | 25.1 | 10.8 | 17.9 | 0 | 13.1 | 32.4 | 25.7 | 25.6 | 31.9 | 22.6 | 31.4 | 14.8 | 15.1 |

**Table A9.** *Cont.*

| Distances (km) | | PCC | | | | | | | | | | | | | |
|---|---|---|---|---|---|---|---|---|---|---|---|---|---|---|---|
| | | 16 | 17 | 18 | 19 | 20 | 21 | 22 | 23 | 24 | 25 | 26 | 27 | 28 | 29 | 30 |
| PCC | 22 | 46 | 34.2 | 33.3 | 26.4 | 26.1 | 13.10 | 0 | 29.3 | 18.9 | 22.9 | 19.4 | 38 | 45.3 | 28.7 | 22.8 |
| | 23 | 25.1 | 36.9 | 19.6 | 40 | 23.9 | 32.40 | 29.30 | 0 | 11.8 | 10.6 | 25.9 | 49.7 | 62.1 | 47.1 | 46 |
| | 24 | 30 | 36.6 | 19.7 | 32.9 | 23.6 | 25.70 | 18.90 | 11.80 | 0 | 9 | 14.3 | 48.7 | 58.6 | 44.8 | 37.2 |
| | 25 | 21.1 | 26.9 | 11 | 30 | 13.9 | 25.60 | 22.90 | 10.60 | 9 | 0 | 23.2 | 38.7 | 50.1 | 36.3 | 38 |
| | 26 | 42.4 | 50.1 | 32.9 | 45 | 37 | 31.9 | 19.4 | 25.9 | 14.3 | 23.2 | 0 | 57.8 | 63.2 | 46.6 | 40.8 |
| | 27 | 38.5 | 14.2 | 33.9 | 15.7 | 26.4 | 22.6 | 38 | 49.7 | 48.7 | 38.7 | 57.8 | 0 | 19.2 | 17.5 | 30.7 |
| | 28 | 57.3 | 33 | 52.7 | 24 | 41.6 | 31.4 | 45.3 | 62.1 | 58.6 | 50.1 | 63.2 | 19.2 | 0 | 19.2 | 31.8 |
| | 29 | 46 | 27 | 35 | 10.3 | 27.9 | 14.8 | 28.7 | 47.1 | 44.8 | 36.3 | 46.6 | 17.5 | 19.2 | 0 | 17.7 |
| | 30 | 55.8 | 36.8 | 40.2 | 20.7 | 33 | 15.1 | 22.8 | 46 | 37.2 | 38 | 40.8 | 30.7 | 31.8 | 17.7 | 0 |

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
