# Peer review of "Extended Maximal Covering Location and Vehicle Routing Problems in Designing Smartphone Waste Collection Channels: A Case Study of Yogyakarta Province, Indonesia"

_sustainability, doi:10.3390/su13168896_

Round 1
Reviewer 1 Report
Dear authors,
I think that your paper does not provide any contribution to the international scientific literature. In addition, in my opinion, the quality of the presentation is very low making the article unreliable for publication in a scientific journal.
Abstract. The statement of novelty is not included. I cannot find the problem that would be solved thanks to your research, neither the future applications for the analysis. How can your research contribute towards sustainable solid waste management? Who can be interested in your research?
The introduction section is very long. It should be halved in order to make it of interest to the readers. It is not well structured, and the logical arrangement of the paragraphs is not clear to me. It happens also in the other sections, where information is scattered and is not provided reliably. It represents a really weak point for a scientific paper. The use of the references in the text is not appropriated. Please, see the guidelines for authors.
Methods: L214-215 I suppose that costs related to smartphone collection are not so high compared with other waste fractions. Why did you choose this type of waste? Why is route optimization required? I think that the study is of low impact, and it does not provide any contribution to support the solid waste management system in Indonesia, which probably requires studies related to bulky waste or hazardous, or municipal waste management. You should better define the reason behind the research as well as the novelty of the study. L222-224 This part is not clear. Where is this information coming from?
Results. L343-344 This part should be reported within the methods section. L346 Units of measure should be better reported. Please, revise the whole text. L345-352 This part should be reported within the methods section.
The use of figures is not appropriated. Figure 1 is not meaningful to the readers. Figure 2 has no sense if the national framework is not provided. Figure 3 is not readable.
Tables are not of interest to the readers and do not report any result.
Discussion. The structure of the text makes it really difficult to follow. While the use of English is fine, the logical structure of the paragraphs and sections is not appropriated. In my opinion, there are too many required for making the paper reliable for publication.
Conclusions do not provide any novel contribution to the scientific literature in order to proceed further towards sustainable solid waste management in developing countries.
Author Response
I think that your paper does not provide any contribution to the international scientific literature. In addition, in my opinion, the quality of the presentation is very low making the article unreliable for publication in a scientific journal.
Response: Thank you for your constructive comments
Point 1: Abstract. The statement of novelty is not included. I cannot find the problem that would be solved thanks to your research, neither the future applications for the analysis. How can your research contribute towards sustainable solid waste management? Who can be interested in your research?
Response 1: The abstract has been improved, including problems, methods, results and conclusions. And do not exceed 200 words. Improvements can be seen in L10-23. In short, in this paper we proposed an extended-Maximum-Covering-Location Problem (e-MCLP) which is the extension of existing MCLP. Classical MCLP does not consider cost, since it is assumed that all investment cost for all areas are the same. However, this is not the case in our problem. Thus, we explicitly include the investment and transportation cost in our model. Detailed explanation regarding this e-MCLP are provided in L247-260
Point 2: The introduction section is very long.
Response 2: We have made the introduction more concise.
Point 2a: It should be halved in order to make it of interest to the readers. It is not well structured, and the logical arrangement of the paragraphs is not clear to me. It happens also in the other sections, where information is scattered and is not provided reliably. It represents a really weak point for a scientific paper.
Response 2a:
- Reasons for smartphone waste as an object of research
Smartphones are electronic products that about 80% of the materials can be recycled effectively [7]. Smartphones contain valuable materials, such as gold, silver, and palladium [8]. Metals in electronic waste, especially smartphones, is higher than primary ore found in the ground. As an illustration, 300-350 grams of secondary gold can be extracted from one ton of cell phones, while every ton of soil in ordinary gold mines only produces 5 grams of primary gold [9]. Resource extraction from e-waste is more economical than extracting metal ores from the ground [10]. So, smartphone recycling is done because the economic benefits outweigh the costs [11]. Proper management of e-waste is necessary to reduce the problem of metal scarcity [8]. The potential for smartphone waste in Indonesia is quite significant. The total population of Indonesia in 2020 is 270,203,917 people [12]. If 63.53% are smartphone users [13], then the total mobile phone users are 171,660,549 people. With the average mobile phone lifetime of 4.7 years [14], and the average weight of a smartphone is 0.5 kg in one year, it produces 36,523,521 units of smartphone waste. When this waste is appropriately managed, in addition to minimizing the environmental impact, it can also provide significant economic benefits by producing 5.48 - 6.39 tons of secondary gold and saving natural resources (L45-60)
- Reasons why it is necessary to determine a collection channel
However, so far, the amount of secondary metal recovered through e-waste recycling has been limited [15]; this is due to the limited supply of e-waste. A preliminary study conducted on smartphone users in Indonesia showed that 59% save non-functioning smartphones, 21% dispose of them, and the rest give them out to other people, sell them, and others. It is because the public does not know what to do with these items. Meanwhile, Yogyakarta is one of the barometer provinces in Indonesia with an improper electronic waste management system through the formal channel. According to previous studies, government drivers are the factor with the most influence on consumers' intentions to participate in smartphone waste collection programs, followed by facility accessibility [16]. This means that the government needs to develop and implement a formal e-waste management system, starting with the e-waste collection process. One of the alternative electronic waste collection programs applicable to Indonesia is the use of DropBox [17], but Yogyakarta province does not currently have any collection points for smartphone waste. Therefore, there is the need to provide a convenient collection channel for the consumers, which is expected to be a major starting point for a formal channel to waste management in the area (L61-L76).
Point 2b: The use of the references in the text is not appropriated. Please, see the guidelines for authors.
Response 2b: We have adjusted the reference to the text according to the guidelines for authors. We've improved the citation of text must be in square brackets
Point 3: Methods
Point 3a: L214-215 I suppose that costs related to smartphone collection are not so high compared with other waste fractions. Why did you choose this type of waste? Why is route optimization required? I think that the study is of low impact, and it does not provide any contribution to support the solid waste management system in Indonesia, which probably requires studies related to bulky waste or hazardous, or municipal waste management. You should better define the reason behind the research as well as the novelty of the study.
Response 3a:
The costs associated with collecting smartphones are not as high as for other large volumes of e-waste, but the developed model can be used for other types of waste. The reason for choosing this type of waste is because it has a higher economic value (containing precious metals such as gold, silver, and palladium) than others, with components that allow up to 80% recycling and a large potential for mobile phone waste. Meanwhile, for now, informal actors dominate the practice of recycling cell phone waste that harms the environment. The collection cost, which is not large, but has high economic and environmental benefits, is expected to motivate both the government to implement the proposed scenario (L185-193)
Transportation routes are needed in this study because smartphones are products with small volumes, so the capacity of the collection center is not as large as vehicle capacity. If one trip only pick-up from one PCC, it becomes inefficient because the vehicle's utility is low, and transportation costs will be higher due to many trips. For this reason, it is necessary to consider the route determination in this study. Routing is expected to increase vehicle utilization and save transportation costs. The output of determining the transportation route is expected to be an input for local governments to schedule waste collection (L197-204)
Point 3b: L222-224 This part is not clear. Where is this information coming from?
Response 3b: Further, a survey conducted on smartphone users, with a total of 325 valid questionnaire showed the consumers is willing to bring their smartphone waste to collection facility with a maximum distance of 11.2 km (L229-231)
Point 4: Results.
Point 4a: L343-344 This part should be reported within the methods section.
Response 4a: We have reported this part in the methods section on L177-178
Point 4b: L346 Units of measure should be better reported. Please, revise the whole text. L345-352 This part should be reported within the methods section.
Response 4b: We have corrected the unit of measure in this section and in the text as a whole. We have reported L205-213 in the methods section
Yogyakarta is one of the provinces in Indonesia located on the Java island with an area of ​​3,178.79 km2. It has a municipality and four regencies including Yogyakarta city as well as Gunung Kidul, Bantul, Sleman, and Kulon Progo regencies with respective areas of 32.5 km2, 1,485.36 km2, 506.85 km2, 574.82 km2, and 579.26 km2. There are 78 districts distributed in these areas with 14, 18, 17, 17, and 12 respectively [70] found in each as indicated in Appendix A. These districts were used as candidates for primary collection centers (PCCs) in this study. The parameters used as input in the mathematical model include the distance between the PCCs, the distance expected by consumers, and the distance from PCC to SCC (L205-213)
Point 4c: The use of figures is not appropriated. Figure 1 is not meaningful to the readers. Figure 2 has no sense if the national framework is not provided. Figure 3 is not readable.
Response 4c: Figure 1 to illustrate the PCC locations selected among the alternative locations. Figure 2 describes the distribution of selected PCC locations in Yogyakarta.Yogyakarta is one of the barometer provinces in Indonesia with an improper electronic waste management system through the formal channel (L66-67). This research is expected to be the initial framework in formulating e-waste management policies in the national formal channel. If this proposal is successfully implemented in Yogyakarta, it is likely to be implemented in other provinces in Indonesia. The developed model can also be used for other solid waste collection scenarios. The proposed e-MCLP model is very suitable for large e-waste because there is no need to proceed to route determination, considering that the supply from PCC may already meet vehicle capacity. One trip only takes in a PCC and then back to SCC again. However, to use the proposed model, it is necessary to consider whether the community is willing to bring their large size/volume e-waste to the provided PCC (L515-523). Figure 3 has been improved, so it’s readable (L421)
Point 4d: Tables are not of interest to the readers and do not report any result.
Response 4d: We have removed table 1 from the text
Point 5: Discussion. The structure of the text makes it really difficult to follow. While the use of English is fine, the logical structure of the paragraphs and sections is not appropriated. In my opinion, there are too many required for making the paper reliable for publication.
Response 5: We have revised the discussion sections (L426-525)
Point 6: Conclusions do not provide any novel contribution to the scientific literature in order to proceed further towards sustainable solid waste management in developing countries.
Response 6: We have improved the conclusion sections (L527-554)
Reviewer 2 Report
Dear Authors,
this research topic is very interesting. This paper is written logically, the text is clear. The asked main question was answered but in my opinion, this article needs to be reworked by the Authors. English language proficiency is also required.
Keywords:
I would rewrite the Keyword. Delete: “nearest neighbor”; instead: “mathematical model”.
Figures and Tables:
I would like to see some data in the form of diagrams (e.g. costs, transport distances).
References:
- DOI numbers are missing.
A list of “Abbreviations” is missing at the end of the article. This would be highly recommended.
Literature references should be enclosed in square brackets in the text.
The “Conclusions” chapter would need to be rewritten by the Authors!
Yours Sincerely
Author Response
Point 1: Dear Authors, this research topic is very interesting. This paper is written logically, the text is clear. The asked main question was answered but in my opinion, this article needs to be reworked by the Authors. English language proficiency is also required.
Response 1: Thank you for your positive comments. We will proofread the manuscript to improve the quality of the English language.
Point 2: Keywords
I would rewrite the Keyword. Delete: “nearest neighbor”; instead: “mathematical model”.
Response 2: I have removed: “nearest neighbors”; instead: "mathematical model". (L25)
Point 3: Figures and Tables:
I would like to see some data in the form of diagrams (e.g. costs, transport distances).
Response 3: The costs involved in the research are only two costs, namely investment costs and transportation costs, information related to costs is conveyed in the text on line L274-279. Dropbox procurement costs are 350,367 USD (1 USD is equivalent to 14,270.75 IDR) with a service life of 5 years; using the straight-line depreciation method, the annual depreciation cost is 70,073.4 USD per dropbox. So that, investment cost per year is 70,073,4 USD per dropbox. The vehicle's fuel consumption is 10 km/liter at 0.67 USD per liter, therefore the transportation cost is 0.067 USD per km.
We present transport distances in the form of distance matrices in appendix B and D
Point 4: References
Point 4a: DOI numbers are missing.
Response 4a: We follow the guidelines for authors
Point 4b: A list of “Abbreviations” is missing at the end of the article. This would be highly recommended.
Response 4b: We have added abbreviations at the end of article (L555)
Point 4c: Literature references should be enclosed in square brackets in the text.
Response 4c: We have corrected the writing of literature references, enclosed in square brackets in the text.
Point 5: The “Conclusions” chapter would need to be rewritten by the Authors!
Response 5: We have rewritten the “Conclusion” chapter (L527-554)
Reviewer 3 Report
The manuscript covers an important topic related to used smartphones, ie collection, transport and the overall management system of this waste in Indonesia. Nevertheless, there were some issues in the article that need to be modified (please see the list below).
- The introduction is quite long, and it lacks the theoretical foundation of the topic - there is no precise information on how many smartphones are used, what kind of waste they are - what they are characterized by, what the current management system for this type of waste looks like, etc. In turn, information on what was done in the research should be included in the next section on methods. Instead, the research goal should be precisely defined and why this topic was taken up. Therefore, I suggest a complete change in the structure of the introduction.
- Figure 1 and Figure 4 are not pasted straight, which looks carelessly.
- The article, but most of all the conclusions, lack information about what actually results from the analyzes carried out - can / do they have an impact on changes in the management system? are they likely to be implemented?
- The entire manuscript should be revised in terms of English editing, for example, some articles are missing. The article should be standardized, e.g. in terms of the font used; also, in my opinion, citations should be in square brackets ([1] instead of (1)) etc.
Author Response
The manuscript covers an important topic related to used smartphones, ie collection, transport and the overall management system of this waste in Indonesia. Nevertheless, there were some issues in the article that need to be modified (please see the list below).
Response: Thank you for your positive comments
Point 1: The introduction is quite long, and it lacks the theoretical foundation of the topic - there is no precise information on how many smartphones are used, what kind of waste they are - what they are characterized by, what the current management system for this type of waste looks like, etc. In turn, information on what was done in the research should be included in the next section on methods. Instead, the research goal should be precisely defined and why this topic was taken up. Therefore, I suggest a complete change in the structure of the introduction.
Response 1: We have improved the introduction section and added information on the reasons for choosing smartphone waste, the potential of the waste, the economic benefits that will be obtained from this waste management, current management system for this type of waste on line L45-76
Smartphones are electronic products that about 80% of the materials can be recycled effectively [7]. Smartphones contain valuable materials, such as gold, silver, and palladium [8]. Metals in electronic waste, especially smartphones, is higher than primary ore found in the ground. As an illustration, 300-350 grams of secondary gold can be extracted from one ton of smartphones, while every ton of soil in ordinary gold mines only produces 5 grams of primary gold [9]. Resource extraction from e-waste is more economical than extracting metal ores from the ground [10]. So, smartphone recycling is done because the economic benefits outweigh the costs [11]. Proper management of e-waste is necessary to reduce the problem of metal scarcity [8]. The potential for smartphone waste in Indonesia is quite significant. The total population of Indonesia in 2020 is 270,203,917 people [12]. If 63.53% are smartphone users [13], then the total smartphone users are 171,660,549 people. With the average smartphone lifetime of 4.7 years [14], and the average weight of a smartphone is 0.5 kg in one year, it produces 36,523,521 units of smartphone waste. When this waste is appropriately managed, in addition to minimizing the environmental impact, it can also provide significant economic benefits by producing 5.48 - 6.39 tons of secondary gold and saving natural resources.
However, so far, the amount of secondary metal recovered through e-waste recycling has been limited [15]; this is due to the limited supply of e-waste. A preliminary study conducted on smartphone users in Indonesia showed that 59% save non-functioning smartphones, 21% dispose of them, and the rest give them out to other people, sell them, and others. It is because the public does not know what to do with these items. Meanwhile, Yogyakarta is one of the barometer provinces in Indonesia with an improper electronic waste management system through the formal channel. According to previous studies, government drivers are the factor with the most influence on consumers' intentions to participate in smartphone waste collection programs, followed by facility accessibility [16]. This means that the government needs to develop and implement a formal e-waste management system, starting with the e-waste collection process. One of the alternative electronic waste collection programs applicable to Indonesia is the use of DropBox [17], but Yogyakarta province does not currently have any collection points for smartphone waste. Therefore, there is the need to provide a convenient collection channel for the consumers, which is expected to be a major starting point for a formal channel to waste management in the area.
Point 2: Figure 1 and Figure 4 are not pasted straight, which looks carelessly.
Response 2: Figure 1 and Figure 4 we have pasted straight (L359 and L452)
Point 3: The article, but most of all the conclusions, lack information about what actually results from the analyzes carried out - can / do they have an impact on changes in the management system? are they likely to be implemented?
Response 3: We have corrected the article, especially the conclusion. We have presented information about what resulted from the analysis carried out, the impact on changes in the management system and the possibilities for implementation
e-MCLP is very suitable for PCCs with large waste volumes because vehicle capacity is filled faster when the volume of waste is large so that there are fewer pick-up points on one route. There are fewer pick-up points in one route; the more routes there will be, the development of this method is suitable for implementation. This model's savings in transportation costs will be felt when the number of routes increases because vehicles will depart and return to SCC more often. That is, the closer CCP distance to the SCC is very beneficial for the vehicle. In this study, the selected location does not affect the investment cost because each candidate location requires a procurement cost of the same amount. However, the developed model can accommodate if each candidate location requires a different investment cost. So later, the selected location will provide a minimum total cost, including investment and transportation costs (L463-474)
Determination of smartphone waste collection routes in the province of Yogyakarta with one route picking up at several PCC points managed to save a mileage of 346.5 km compared to one route only picking up at one PCC point a total of 30 pick-up points. If the smartphone waste collection is done once a week, this shorter distance can provide transportation cost savings of 2,214.39 USD per year. The area of Yogyakarta province is only 0.16% of the territory of Indonesia; if this model is implemented nationally, the perceived transportation cost savings will be more than 1 million USD (L508-514)
This research is expected to be the initial framework in formulating e-waste management policies in the national formal channel. If this proposal is successfully implemented in Yogyakarta, it is likely to be implemented in other provinces in Indonesia. The developed model can also be used for other solid waste collection scenarios. The proposed e-MCLP model is very suitable for large e-waste because there is no need to proceed to route determination, considering that the supply from PCC may already meet vehicle capacity. One trip only takes in a PCC and then back to SCC again. However, to use the proposed model, it is necessary to consider whether the community is willing to bring their large size/volume e-waste to the provided PCC. This research is also the first step in waste management, especially electronic waste, which will then be followed by the next stage of management, which includes separation, repair, recycling, remanufacturing, or disposal. (L515-525)
Point 4: The entire manuscript should be revised in terms of English editing, for example, some articles are missing. The article should be standardized, e.g. in terms of the font used; also, in my opinion, citations should be in square brackets ([1] instead of (1)) etc.
Response 4: We will proofread the manuscript to improve the quality of the English language. We have revised in the whole text, the citation of text must be in square brackets ([1] instead of (1)), the unit of measure and the writing of numbers. We have also checked the consistency of the font used in both type and size.
Round 2
Reviewer 2 Report
Dear Authors,
You accepted my suggestions and rewrote the article based on my suggestions. Congratulations!
Yours Sincerely
Reviewer 3 Report
The authors made the necessary corrections that significantly improved the quality of the article. In my opinion, the introduction could be better prepared, but the current general reception of the article is sufficient.